# Impact of Intersectoral Dependencies in National Production on Wastewater Discharges: An Extended Input–Output Study of the Croatian Economy

**Saša Čegar** [1,*], **Nada Denona Bogović** [1] **and Alen Jugović** [2,*]

1. Faculty of Economics and Business, University of Rijeka, Ivana Filipovića 4, 51 000 Rijeka, Croatia; nada.denona.bogovic@efri.hr
2. Faculty of Maritime Studies, University of Rijeka, Studentska ulica 2, 51 000 Rijeka, Croatia
* Correspondence: sasa.cegar@efri.hr (S.Č.); ajugovic@pfri.hr (A.J.)

**Abstract:** The Croatian economy performs unfavorably in terms of the impact of production on wastewater discharges, which is particularly pronounced in the industrial sectors. Each unit of gross industrial value added produced in Croatia generates significantly more wastewater discharges than in most European countries with a similar level of economic development. Moreover, in 2020, only 26.9% of the total industrial wastewater discharges of the Croatian economy were treated, while 76.2% of the total industrial wastewater discharges were directly discharged into the environment. Since most of the industrial production in the Croatian economy is destined to meet the intermediate needs of other sectors, policy makers in Croatia must take into account that the level of industrial wastewater discharges is also influenced by the production level of sectors that depend on the intermediate products of wastewater-intensive industries. For this reason, we developed a wastewater extended input–output model of the Croatian economy to determine and analyze the impact of intersectoral linkages in Croatian production systems on the amount of untreated wastewater discharges. The results of the study show that wastewater flows in the Croatian economy are largely generated by the processes of production and consumption of intermediate products from the chemical and petroleum refining sectors, which also account for the largest share of the calculated wastewater footprint of total Croatian production. In light of the emerging empirical evidence, it can be concluded that targeting market-based and regulation-based measures at wastewater-intensive producers is not sufficient to reduce the relatively high level of untreated wastewater discharges in the Croatian economy. There is also a need for appropriate integrated policy measures in sectors that have a large wastewater footprint due to their established supply chains.

**Keywords:** Croatian economy; EEIO analysis; indirect wastewater intensity; cumulative wastewater intensity; wastewater footprint; forward and backward linkages





## 1. Introduction

Wastewater streams discharged into water reservoirs are of great concern as they contribute the most to inorganic, organic and thermal pollution of water resources [1]. By definition, wastewater is water used in households, industries or other economic and social facilities that does not serve a useful purpose or does not meet physical, chemical, biological or other water quality standards without adequate treatment [2] (p. 7). Since raw wastewater streams contain pollutants, bacteria and viruses that can seriously damage ecosystems and endanger human health, adequate wastewater treatment prior to discharge into the environment is an important requirement for maintaining and improving the quality of available water resources [3]. Domestic wastewater, service wastewater and manufacturing wastewater are critical types of wastewater, of which manufacturing wastewater is more difficult to treat due to its complex pollution components and high concentration of pollution [4]. In order to reduce overall wastewater discharges, it is particularly important

to have a comprehensive understanding of how they are caused by production activities in the economy [5,6].

Wastewater generated in the economy is the result of the material and energetic residues of production. In productions where water is used as a medium for technological processes or for other purposes (e.g., cooling and cleaning of production equipment), unused production residues in the form of solid, liquid or gaseous pollutants are dissolved or simply remain in the water. Since the discharge of polluted water into the environment negatively affects the characteristics of the water bodies that receive the wastewater, increasing production in wastewater-intensive sectors further threatens the quality of water resources [7]. However, identifying and monitoring sectors that directly pollute the environment with untreated wastewater (hereafter wastewater) is not sufficient to fully understand the incidence of total wastewater discharges in the national economy. In fact, many studies have shown that wastewater pollution is transferred indirectly, i.e., virtually, through economic activities, where virtual wastewater flows imply wastewater embodied in intermediate and final products and services [8]. Understanding the mechanisms of virtual wastewater transfer through economic activities is a complex research task. This is because wastewater-intensive sectors are linked to other sectors in the economy's production system through a network of supply chains. The sectors involved in the processes of production and distribution of the outputs of the wastewater-intensive sectors therefore indirectly influence the total wastewater discharge in the economy. Considering that certain sectors discharge large amounts of wastewater directly into the environment to meet the production needs of other sectors [9], an appropriate method of virtual wastewater accounting plays an important role in the development of integrated wastewater policies.

When it comes to the relevance of using certain methodological procedures for accounting and analyzing virtual wastewater flows in economies, the most frequently mentioned in the literature is the environmentally extended input–output analysis (EEIO). EEIO analysis is a long-established input–output technique for quantifying environmental impacts that occur along supply chains and analyzing their linkages to economic activities. Unlike bottom-up methods that rely on more detailed data for environmental accounting, EEIO analysis is a top-down method that can distinguish direct and indirect environmental impacts caused by intermediate and final demand [10]. EEIO analysis is based on a system of linear equations that describe the established intersectoral linkages in the production structure of a given economy, including its links to sources of primary inputs and final consumption. Assuming that the economic structure and relative prices are fixed, EEIO models allow the calculation of the cumulative environmental impacts of meeting final demand for a given sector's output, regardless of the degree of complexity of the supply chains that support that output. Because of their mathematical simplicity and clarity in interpreting the results obtained, EEIO models have become a powerful tool in research on the relationship between the economy and the environment and in information-based policy making [11–13].

Given the strategic importance of Croatia's renewable water resources [14,15], it is particularly important for policy makers to understand the key drivers and impacts of virtual wastewater flows in the Croatian national economy. According to Aquastat [16], Croatia has 25,222 m$^3$ of renewable water resources per capita per year, ranking fourth in Europe and 31st in the world. Due to the low population density and modest water demand for industry and agriculture, only 2% of the available renewable water resources are withdrawn annually in Croatia. Along with the relative abundance of renewable water resources, most of the water bodies on Croatian territory are of good quality [17].

However, due to insufficient development of public wastewater collection, treatment and disposal systems, there is a significant risk for maintaining and improving the quality status of water resources in Croatia. Therefore, with the support of EU funds, Croatia will invest approximately USD 2.6 billion over the next three to five years in the construction and development of infrastructure needed to improve water and wastewater services within the national system boundary [18]. Since the access of households and producers

to the public wastewater system is very low, the investment cycle will focus mainly on the development of wastewater systems. Currently, centralized wastewater systems are built only in large urban and industrial centers, so the capacity of industrial wastewater treatment in the various production sectors of the Croatian economy is not at the same level of development. Large companies generally have better wastewater treatment infrastructure, but the majority of wastewater producers still lack modern equipment to reduce and properly treat their wastewater discharges [19] (p. 37). As a result of underdeveloped industrial wastewater systems and low capacity of industrial wastewater treatment, only 26.9% of total industrial wastewater discharges in the Croatian economy were treated in 2020, while 76.2% of total industrial wastewater discharges were discharged directly into natural receiving environmental elements—watercourses, sea, soil, lakes and reservoirs [20]. Moreover, Croatian industry has a relatively low value added per unit of untreated wastewater discharge compared to industry in European countries for which recent Eurostat data on wastewater discharges were available (see Figure 1) [21,22].

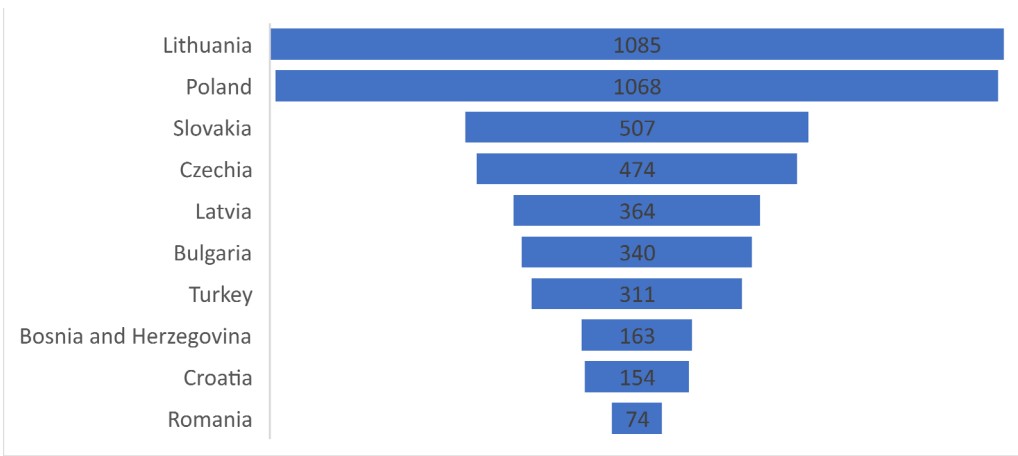

**Figure 1.** Industry gross value added in current prices per m$^3$ of untreated industrial wastewater discharge in selected European countries, 2018 (authors' calculation based on Eurostat data, see Table A1).

Although industry is the largest contributor of wastewater in EU economies [21], the values presented in Figure 1 show that industrial production in Croatia is more wastewater-intensive than in countries with a similar level of economic development. This means that each unit of gross industrial value added produced in Croatia generates more wastewater discharges than in comparable countries. However, the wastewater productivity or intensity indicators do not provide a complete picture of the nature of the impact of different production sectors on wastewater discharges in the Croatian economy. This is mainly because a significant part of industrial production is consumed as intermediate products in other sectors [23], so the level of industrial wastewater discharges in the Croatian economy is determined by the production level of other sectors. Therefore, when considering an appropriate approach to wastewater reduction in the Croatian economy, it must be taken into account that part of the responsibility lies with the sectors involved in the intermediate processes of production and consumption of wastewater-intensive goods and services. For this very reason, it is necessary to better understand the interconnectedness of industry and other sectors within the Croatian production system in terms of cross-sectoral virtual wastewater flows. This is also the main objective of this study, and our empirical strategy is presented in Figure 2.

| WASTEWATER AND ECONOMIC DATA | |
|---|---|
| Direct untreated wastewater discharges of Croatian production sectors | Input-output table of the Croatian economy |

⇩

| WASTEWATER EXTENDED INPUT-OUTPUT MODEL OF THE CROATIAN ECONOMY | |
|---|---|
| Calculation of indicators of wastewater intensity of sectoral production:<br>- *Direct wastewater intensity* (measures how much wastewater a given sector must discharge to produce one unit of its output).<br>- *Indirect wastewater intensity* (measures how much wastewater the economy must discharge to produce inputs needed for the production of one unit of output in a given sector).<br>-*Cumulative wastewater intensity* (measures the amount of all wastewater discharges in the economy that are directly and indirectly associated with the production of one unit of output in a given sector) | Calculation of wastewater multipliers of sectoral production:<br>- *Direct wastewater multipliers* (measures the relative increase in indirect wastewater discharge in relation to the increase in the sector's direct wastewater discharge).<br>- *Cumulative wastewater multipliers* (measures the relative increase in cumulative wastewater discharge (direct+indirect) associated with the increase in the sector's direct wastewater discharge). |
| Calculation of wastewater footprint of production sectors:<br>- *Domestic wastewater footprint* (difference between total wastewater discharges associated with the production of goods and services for domestic consumption and total wastewater discharges associated with the production of exported goods and services).<br>- *Net imported wastewater footprint* (difference between total wastewater discharges associated with the production of imported and exported goods and services).<br>- *Total wastewater footprint* (sum of domestic and net imported wastewater footprint) | Calculation of the relative strength of sectoral forward and backward linkages with respect to wastewater discharge:<br>- *Pull index* (measures the relative impact of a change in final demand for products in a given sector on total wastewater generation in all productive sectors of the economy).<br>- *Push index* (measures the relative impact of a change in aggregate final demand for products across all production sectors in the economy on wastewater discharge in a given sector) |

⇩

| NEWLY OBTAINED EMPIRICAL RESULTS | |
|---|---|
| Identification of sectors that directly, indirectly and cumulatively play a key role in the generation of wastewater discharges in the Croatian economy | Integrated policy recommendations for reducing total wastewater discharges in the Croatian economy |

**Figure 2.** Empirical strategy of the study.

Based on an extended input–output model for wastewater, we calculate and analyze the cumulative, direct and indirect wastewater intensity of the different production sectors in the Croatian economy, as well as their multiplicative impact on the discharge of untreated wastewater. We also calculate and analyze the sectoral wastewater footprints in the Croatian economy and the relative strength of sectoral forward and backward linkages in generating virtual wastewater flows. Taking into account the peculiarities of the main objective of

the research and the methodology applied, the results of this study, presented below, are elaborated in four main sections. The literature review section provides an overview of the wastewater extended input–output studies, based on which empirical gaps in the literature on the Croatian economy are identified. The second section explains step by step the development of the used input–output model for the Croatian economy and the model data. In the empirical section, the newly obtained results are analyzed, interpreted and discussed. Finally, the concluding section summarizes the main findings and empirical results of the study and provides policy suggestions for minimizing wastewater discharge in the Croatian economy. At the end of the conclusion, the main limitations of the study are highlighted and recommendations for future research are provided.

## 2. Overview of the Literature

Wastewater pollution is considered one of the most serious global environmental problems, as pollution of water sources can limit and prohibit different types of water use [24,25]. In most countries, the majority of wastewater is discharged directly into the environment without adequate treatment, negatively affecting human health, economic productivity, the quality of surrounding freshwater resources and ecosystems. If current trends continue, water quality will continue to deteriorate in the coming decades, contributing to water stress and limiting sustainable economic development [26]. Therefore, the interdisciplinary field of wastewater pollution research is of great importance for mitigating wastewater pollution, conserving water resources and protecting water-related ecosystems [27–29], which is critical for achieving a balance between economic and environmental sustainability [30,31].

It is well known that input–output models are particularly well suited to fully describe the complexity of intersectoral relationships in and across economies and to estimate how changes in the components of final demand and various primary inputs affect different economic, social or environmental variables. Although certain forms and segments of the analysis of intersectoral relations can be found in the earlier literature [32], there is a general consensus in the scientific and academic community that the main founder of input–output analysis is Professor Wassily Leontief, who was awarded the Nobel Prize in 1973 for his long-standing contribution to the study of input–output analysis. Since traditional input–output models were theoretically extended in the 1960s to account for the feedback between the economy and the environment by adding physical information (in rows or columns) to monetary input–output tables [33–35], input–output analysis has become a widely applicable empirical tool in the field of interdisciplinary environmental studies [36]. EEIO models allow the quantification of direct and indirect resource use and environmental pollution associated with meeting final demand. Therefore, these models provide a universal methodological basis for identifying the key drivers and sources of the environmental pressures embodied in demand supply chains [37].

Since the Statistical Office of UN published a manual on the System of Integrated Environmental and Economic Accounts (SEEA) in 1993 [38]—which elaborated methodological guidelines and examples for linking environmental statistics, including water statistics, to the system of national accounts—statistical offices in many countries have begun to use economic standards and classifications in the collection, organization and publication of data on water issues. To date, the SEEA framework has been revised and expanded several times and now includes a number of subsystems specifically designed to capture and monitor typical areas of interaction between the economy and the environment. The System of Environmental-Economic Accounts for Water (SEEA-Water) was published in 2012 as the first international integrated accounting standard for water [39]. This spurred the adoption and development of integrated water accounting in many countries and set the stage for more frequent use of input–output analysis in the study of virtual water, water pollution and wastewater flows within and across economies at different territorial scales [40].

Through the application of the EEIO method, numerous scientists have uncovered a wide range of wastewater discharge problems. For example, in the context of river pollution in Indonesia, a study published in 2003 identified pollution control sectors and pollution

prevention sectors [41]. The study concluded that sectors with high parameters for effective pollution control should be given high incentives to eliminate their pollution as much as possible because it is relatively inexpensive to do so (e.g., subsidies for pollution control and strict control of wastewater discharge, high progressive wastewater tax). Similarly, consumers should be strongly discouraged (e.g., by a relatively high sales tax) from increasing their demand for products from sectors with high effective pollution control parameters, since increasing demand for these sectors results in relatively large amounts of river water pollution that is relatively expensive to clean up.

Another study in 2005 analyzed the relationships between production processes and water pollution based on satellite water accounts and input–output tables for the Spanish economy [42]. The objective was to identify the role of different sectoral blocks as producers and consumers of different types of pollution and to examine how water pollution responds to changes in final demand. A similar research approach was used in a study of the Indian economy in 2012 [43]. By extending the traditional input–output model with a matrix of coefficients for various pollutants in wastewater discharges, the study analyzed the intensity and pattern of direct and indirect pollution of individual industries in the Indian economy. It was found that the production of textiles, leather goods, paper products and chemicals cumulatively contribute the most to water pollution in the Indian economy, while among final demand categories, private consumption expenditure has the largest impact on almost all pollutants.

Based on the assumption that it is important for sustainable river basin development to consider not only direct but also indirect impacts on water demand and pollution, a 2013 study from China applied a hybrid input–output model to analyze water consumption and wastewater discharge in the Haihe River Basin [44]. The results of the study show that water consumption and pollution can be reduced if the structure of production, consumption and trade in the Haihe River Basin is adjusted to its carrying capacity.

To determine the dynamics of water pollution in an international context, a group of authors in 2017 analyzed the main global drivers of graywater discharge and efforts to reduce graywater [45]. Using the structural decomposition analysis method and data from the World Input–output Database, the authors assessed the impact of changes in intermediate industrial flows, domestic final demand, exports and water demand on graywater production. In their study, the authors concluded that increases in income and population in developing countries put pressure on graywater production in primary sectors to meet increasing population demands. On the other hand, industries that are significant graywater producers are being relocated from developed countries to developing countries, where inadequate technological development and pollution regulation are leading to even higher graywater production than before. Therefore, secondary sector policies must pay special attention to key industries in global value chains that contribute most to water pollution.

The most recent EEIO study on direct and indirect relationships between water pollution and production of different sectors was conducted in 2020 using the Nepalese economy as an example [46]. The study identified key water pollution control industries (i.e., heavy polluters that incur relatively low cleanup costs) and water pollution prevention industries (i.e., polluters that accelerate pollution by inducing the release of pollutants from other related sectors). The results suggest that incentives to comply with wastewater standards are more effective for industries that reduce pollution, while penalties are more effective for industries that avoid pollution because they support Nepal's economy.

In order to systematically analyze wastewater discharge in the process of industrialization and urbanization of Guangdong Province in China, a group of authors developed a dynamic wastewater-induced input–output model in 2020 [47]. By combining input–output analysis, ecological network analysis and structural decomposition analysis, the authors found that paper, computer and machinery manufacturing and services are the key industries responsible for large amounts of wastewater discharges and unhealthy source–discharge relationships. In addition, final demand was found to be the largest driver of wastewater discharge.

Summarizing the literature on wastewater- and water pollution-based EEIO analyses, one can conclude that the intensity of the impact of individual production sectors on total wastewater and pollution discharge depends primarily on the complexity of their supply chains and the importance of wastewater-intensive production in the national or regional economies to which they belong. Therefore, empirical results vary from country to country. As far as the authors of this study are aware, no empirical study on input–output analysis of wastewater flows or water pollution flows in the Croatian economy has been published at the time of writing. Previous studies have relied exclusively on analyses of direct wastewater discharges, mostly as part of the development of the national strategy and key planning documents for water protection and management in Croatia [48]. To date, only two published studies from 2020 have conducted an EEIO analysis of sectoral water consumption in the Croatian economy [49,50]. Therefore, this study develops the first wastewater-based EEIO model for the Croatian economy.

## 3. Methodological Framework

In this section, the wastewater extended input–output model of the Croatian economy is presented and the origin of its indicators is explained. It also critically examines the assumptions and limitations of using this model as an analytical and simulation tool and describes the data sources used in the model.

### 3.1. Basic Input–Output Model

The following explanation of the basic input–output model is based on various literature sources ([51] (p. 274), [52] (pp. 22–27), [53] (pp. 14–23), [54] (pp. 486–489), [55] (pp. 10–34)).

When the national economy is divided into $n$ sectors, the structural relationships in the production and distribution of the total annual value of national output, as well as the relationship between the total output of each sector and its total direct wastewater discharge, can be illustrated as in Figure 3.

| Output \ Input | | Intermediate use (demand) | | | | | Final use (demand) | Total outputs |
|---|---|---|---|---|---|---|---|---|
| | Sectors | 1 | ... | j | ... | n | | |
| Intermediate inputs (supply) | 1 | $x_{11}$ | ... | $x_{1j}$ | ... | $x_{1n}$ | $f_1$ | $X_1$ |
| | ⋮ | ⋮ | | ⋮ | | ⋮ | ⋮ | ⋮ |
| | I | $x_{i1}$ | ... | $x_{ij}$ | ... | $x_{in}$ | $f_i$ | $X_i$ |
| | ⋮ | ⋮ | | ⋮ | | ⋮ | ⋮ | ⋮ |
| | N | $x_{n1}$ | ... | $x_{nj}$ | ... | $x_{nn}$ | $f_n$ | $X_n$ |
| Primary inputs (supply) | | $p_1$ | ... | $p_j$ | ... | $p_n$ | | |
| Total inputs | | $X_1$ | ... | $X_j$ | ... | $X_n$ | | |
| Wastewater discharges | | $w_1$ | ... | $w_j$ | ... | $w_n$ | | |

**Figure 3.** Wastewater extended input–output table.

Since in the biproportional matrix of intermediate consumption, sector $i$ supplies part of its output to other sectors and sector $j$ uses it as a production input, $x_{ij}$ denotes the part of the value of sector $i$'s annual output that goes into sector $j$'s production. Thus, if $X_i$ is the gross output value of sector $i$ and $f_i$ is the total final use of sector $i$'s output, then the total annual output of sector $i$ from a demand perspective can be defined by the following equation:

$$X_i = \sum_{j=1}^{n} x_{ij} + f_i, \text{ for each } i = 1, 2, \ldots, n \qquad (1)$$

The intensity of direct intersectoral dependencies in production processes is expressed by the technical coefficients of production, which are calculated as follows:

$$a_{ij} = \frac{x_{ij}}{X_j} \qquad (2)$$

Since the technical coefficient $a_{ij}$ reflects the value of sector $i$'s output expended by sector $j$ to produce one unit of its own output, sector $i$'s share in sector $j$'s total intermediate consumption can be replaced by the function $x_{ij} = a_{ij}X_j$. Accordingly, the system of Equation (1) can be transformed into an equivalent system:

$$X_i = \sum_{j=1}^{n} a_{ij}X_j + f_i, \text{ for each } i = 1, 2, \ldots, n \qquad (3)$$

When $a_{ij}$ are constants and $f_1, \ldots, f_n$ are given values, the newly obtained system of Equation (3) consists essentially of $n$ linear equations with $n$ unknowns describing the direct relationship between the outputs of all sectors into which the economy is divided. Calculating the multiplicative or cumulative effect of the unit change in the output of each sector on the level of the output of the other sectors therefore implies solving the mathematical problem of finding new values $X_1, \ldots, X_n$ resulting from the change in the value of any $f_i$. Accordingly, this problem can be solved with the help of matrices:

$$x = Ax + f \qquad (4)$$

where:

- x is the vector column of the total output ($X_i$);
- A is the matrix ($n \times n$) of the technical coefficients ($a_{ij}$); and
- f is the vector column of final demand ($f_i$).

The solution of Equation (4) is in fact *the basic Leontief input–output model of production,* linking the total output of each sector and the final demand for the output of each sector in the economy:

$$x = (I - A)^{-1}f \qquad (5)$$

where:

- I is unit matrix ($n \times n$); and
- $(I - A)^{-1}$ is the inverse Leontief matrix (L, $n \times n$).

Thus, the elements of the matrix L($l_{ij}$) measure the cumulative (direct and indirect) impact of the unit change in the level of demand for final products and services of sector $j$ on the total output of sector $i$.

$$l_{ij} = \frac{\partial X_i}{\partial f_j} \qquad (6)$$

When interpreting the results obtained on the basis of the derived model, the following assumptions must be taken into account ([56] (p. 4), [57] (pp. 635–637)):

1. All sectors are characterized by perfect homogeneity of production, which means that all firms within a given sector have the same technological production base and produce exactly the same products;
2. All sectoral inputs are perfect complements in production and cannot be substituted, i.e., the production structure of all sectors is fixed and therefore they have constant returns in the volume of production; and
3. The production capacities of all sectors are absolutely sufficient, which allows a proportional increase in their production in relation to the increase in final demand.

Although the assumptions of *Leontief's input–output model* do not accurately reflect how the production systems of economies operate in reality, the model can still be a very useful tool for analyzing sectoral interdependencies in the economy. This is mainly because the technical structure the production system of the economy in a given period is authentically

reflected by the A-matrix, which allows the calculation and analysis of direct and indirect linkages within the economy as a complex system. It is a fact that the sudden emergence of new activities in the economy, changes in relative prices or, for example, changes in the size and structure of the capital stock can lead to significant changes in the technical basis of the production processes in the economy, but in reality, the technical coefficients generally do not change so rapidly [58,59]. Thus, different variants of this model can also be useful for simulations and forecasts. However, they are not used to precisely quantify the multiplicative effects of endogenous or exogenous factors on the production system of the economy, but to determine the direction and magnitude of these effects. Therefore, extending the model to include wastewater data can definitely help us to better understand the impact of intersectoral production dependencies on wastewater discharges.

*3.2. Wastewater Extended Input–Output Model and Wastewater Multipliers*

The derivation and interpretation of the wastewater extended input–output model presented below is based on several different literature sources [9,60–62].

Although discharged wastewaters are residues of production processes, they can be interpreted in technological terms as the amount of environmental impact required to produce one unit of a given product. Therefore, wastewater discharges are presented as production inputs in Figure 3. In this way, the coefficient of direct wastewater discharge can be calculated, i.e., the coefficient of direct wastewater intensity of sector $j$ ($d_j$):

$$d_j = \frac{w_j}{X_j} \tag{7}$$

where:

- $w_j$ is the total direct wastewater discharge from sector $j$; and
- $X_j$ is the value of the total input to sector $j$.

In the extended version of the model, the diagonal matrix of order $n$ ($\hat{d}$) contains the coefficients of direct wastewater discharge. Thus, by multiplying the matrix $\hat{d}$ by the matrix L, one can specify, for any given sector, how much wastewater must be discharged throughout the economy to produce one unit of sectoral output:

$$T = \hat{d}(I - A)^{-1} \tag{8}$$

The elements of the square matrix T ($t_{ij}$) of order $n$ measure the cumulative wastewater discharges of sector $i$ that are directly and indirectly associated with the production of one unit of output in sector $j$. The cumulative wastewater intensity of sector $j$ is therefore calculated by summing all the elements in the $j$th column of the matrix. Considering the flows of virtual wastewater under the intermediate supply of sector $i$, the cumulative wastewater discharge of sector $i$ required for all sectors to produce one unit of output is calculated as the sum of the elements in the $i$th row of matrix T.

$$m_j = \frac{t_j}{d_j} \tag{9}$$

$$m_j^{ind} = m_j^d - 1 \tag{10}$$

The multiplier $m_j$ indicates the increase in sector $j$'s cumulative wastewater discharge associated with the increase in its direct wastewater discharge by 1 m$^3$. Similarly, the multiplier $m_j^{ind}$ indicates the increase in sector $j$'s indirect wastewater discharge associated with the increase in its direct wastewater discharge by 1 m$^3$.

The matrix of intersectoral wastewater flows (W) is obtained by subtracting the matrix of direct wastewater discharge coefficients ($\hat{d}$) from the cumulative wastewater intensity matrix (T):

$$W = T - \hat{d} \tag{11}$$

The columns of the square matrix W of order *n* reflect the structure of indirect wastewater discharge associated with the unit of production of each sector. Therefore, the total indirect wastewater discharge of sector *j* per its unit of production is the sum of all elements in the *j*th column of matrix W ($w_{ij}^{inter}$).

Based on the elements of the matrix W, a matrix of the technical coefficients of indirect wastewater discharge Q ($q_{ij}$) can be derived:

$$q_{ij} = \frac{w_{ij}^{inter}}{d_j} \tag{12}$$

The technical coefficient $q_{ij}$ indicates the amount of cumulative wastewater discharge of sector *i* that relates to 1 m$^3$ of direct wastewater discharge of sector *j*. Thus, by summing the elements in the *j*th column of matrix Q, a multiplier of indirect wastewater discharge is obtained (see Equation (10)).

*3.3. Wastewater Footprint*

The wastewater footprint is a more complex indicator compared to the previously derived indicators for direct, indirect and cumulative wastewater discharges. This is because it also includes international virtual wastewater flows that are included in foreign trade exchanges [63,64]. In this context, the import of virtual wastewater represents a part of the total wastewater footprint of the importing economy, as it is an indirect impact on the environment abroad. In contrast, the export of virtual wastewater is an indirect impact of foreign economies on the domestic environment and therefore does not contribute to the total wastewater footprint of the exporting economy. This means that the total wastewater footprint of the economy's production sectors in a given year can be defined as the sum of the volume of their total annual domestic wastewater discharges (i.e., the domestic wastewater footprint) and the volume of embodied virtual wastewater in their total annual net import (i.e., the net imported wastewater footprint).

Since it is possible to quantify the direct and indirect environmental impacts of foreign trade within the EEIO model, the extended input–output model for wastewater presented earlier allows the calculation of the imported, exported, domestic and total wastewater footprint of all production sectors in a given economy.

The domestic wastewater footprint (DWF) is the difference between the total amount of wastewater discharged to the environment by domestic producers due to production for domestic consumption and the total amount of wastewater discharged to the environment by domestic producers due to exported production. Domestic producers are natural or legal persons established in the territory of Croatia who produce goods and services in this territory in raw, semi-processed or processed form:

$$\text{DWF} = \hat{d}_1 \, (\text{I} - \text{A})^{-1} \hat{f}_{dom} \tag{13}$$

where:

- $\hat{d}_1$ is the diagonal matrix of the coefficients for direct wastewater discharge;
- $(\text{I} - \text{A})^{-1}$ is the inverse Leontief matrix; and
- $\hat{f}_{dom}$ is the diagonal matrix of domestic demand (domestic demand = the value of final demand—the value of exports, see Figure A1).

Given the limitations of the EEIO model, the imported wastewater footprint calculation assumes that domestic producers have the same intensity of direct and indirect wastewater discharges as foreign producers [65]. It is important to emphasize that this assumption differs considerably from reality, since different economies, especially those that differ in their development and international competitiveness, are characterized by different technological equipment of production and, consequently, by different intensity of wastewater discharge on sector and company levels. Nevertheless, the imported wastewater footprint calculated in this way has analytical value because it indicates the extent

to which the environment would be further polluted by wastewater discharges from the national economy if some of the imports were replaced by domestic production. Under this assumption, the net imported wastewater footprint (NIWF) can be calculated as follows:

$$\text{NIWF} = \hat{d}_1 \, (\text{I} - \text{A})^{-1}(\hat{x} - \hat{m}) \tag{14}$$

where $\hat{x}$ and $\hat{m}$ are diagonal matrices of exports and imports.

Finally, the total wastewater footprint (TWF) can be derived by adding the footprints of domestic and net imported wastewater:

$$\text{TWF} = \text{DWF} + \text{NIWF} \tag{15}$$

*3.4. Pull and Push Wastewater Discharge Indices*

The relative strength of cross-sectoral linkages of production sectors in a given economy can be measured with so-called pull and push indices. From the perspective of wastewater discharges, backward linkages mean the effect of a change in final demand for sector $j$'s output on wastewater discharges within sectors that directly or indirectly serve sector $j$'s demand for intermediate products and services. Forward linkages, on the other hand, are the extent to which wastewater discharges in sector $i$ respond to changes in demand for final products and services from sectors whose production depends directly or indirectly on intermediate supplies from sector $i$ [66] (p. 1143).

To calculate the relative strength of cross-sectoral linkages, one must first determine their absolute strength. The value of the absolute strength of the backward linkage of sector $j$ ($B_j$) is equal to the sum of the elements in the $j$th column of the matrix of cumulative wastewater discharge coefficients (T) (see Equation (8)):

$$B_j = \sum_{i=1}^{n} t_{ij} \tag{16}$$

To calculate the absolute strength of forward linkage, one must use an alternative supply-driven input–output model, called *the Ghosh input–output model* by its author. Unlike Leontief's demand-driven model, Ghosh's model links total output (X) and primary inputs ($p$) [67,68]:

$$\text{x} = (\text{I} - \text{B})^{-1}p \tag{17}$$

where:

- B is the matrix ($n \times n$) of technical allocation coefficients ($b_{ij} = x_{ij}/X_i$); and
- $(\text{I} - \text{B})^{-1}$ is the inverse matrix of Ghosh (G, $n \times n$).

The values in matrix G ($g_{ij}$) quantify the total effect of the unit change in sector $i$'s primary inputs on sector $j$'s total output. Consequently, the absolute strength of sector $i$'s forward linkage with respect to wastewater discharge ($F_i$) is calculated by summing the elements in the ith row of the matrix of cumulative wastewater supply coefficients (V). The matrix V is a product of the Ghosh inverse and the diagonal matrix of coefficients for direct wastewater discharge ($\hat{d}$):

$$\text{V} = \hat{d}(I - B)^{-1} \tag{18}$$

$$F_i = \sum_{j=1}^{n} v_{ij} \tag{19}$$

As mentioned at the beginning of this section, the relative strength of cross-sectoral linkages in a given economy is measured by pull and push indices, which essentially represent the deviations of sectoral forward and backward linkages relative to their average value at the level of all production sectors within the observed economy [61,66,69–71]:

$$PLI_j = \frac{B_j}{\frac{1}{n}\sum_{j=1}^{n} B_j} \tag{20}$$

$$PSI_i = \frac{F_i}{\frac{1}{n}\sum_{i=1}^{n} F_i} \tag{21}$$

If sector $j$ has a pull index ($PLI_j$) value greater than 1, a unit increase/decrease in demand for its final products and services leads to an above-average increase/decrease in wastewater discharge in all domestic production sectors. In contrast, a sector $i$ push index value greater than 1 indicates that a unit increase/decrease in the demand for final products and services of all domestic production sectors leads to an above-average increase/decrease in wastewater discharge in sector $i$.

*3.5. Data Sources*

The EEIO model used in this study is based on the latest official input–output table for the Croatian economy for 2010, published in 2015 [72] (see Table A1). In order to align the monetary input–output data with the available data on wastewater discharges, the original $65 \times 65$ version of the input–output table was reduced to a $24 \times 24$ format, i.e., the national production system was divided into 24 sectors (see Table 1).

**Table 1.** Classification of production sectors.

| Labels | Sectors | Labels | Sectors |
|:---:|:---:|:---:|:---:|
| 1 | Primary sector (agriculture, forestry and fisheries) | 13 | Metallic sector |
| 2 | Mining and quarrying | 14 | Manufacture of computer, electronic and optical products and electrical equipment |
| 3 | Food sector [1] | 15 | Manufacture of machinery and equipment |
| 4 | Textile sector [2] | 16 | Manufacture of motor vehicles, trailers, semi-trailers and their transport equipment |
| 5 | Wood-processing sector [3] | 17 | Manufacturing of furniture and other manufacturing |
| 6 | Manufacture of paper and paper products | 18 | Machinery and equipment repair and installation |
| 7 | Printing and reproduction of recorded media | 19 | Power sector [5] |
| 8 | Petroleum refining sector [4] | 20 | Water supply [6] |
| 9 | Chemical sector | 21 | Environmental and waste sector [7] |
| 10 | Pharmaceutical sector | 22 | Construction |
| 11 | Manufacture of rubber and plastic products | 23 | Hospitality sector |
| 12 | Non-metallic sector | 24 | Other services |

Notes: [1] Includes also the manufacture of beverages and tobacco products. [2] Includes also the manufacture of wearing apparel and leather products. [3] Manufacture of wood and of products of wood and cork, except furniture; manufacture of articles of straw and plaiting materials. [4] Includes also the manufacture of coke. [5] Supply of electricity, gas, steam and air-conditioning. [6] Includes also water collection and treatment. [7] Sewerage, waste management and environmental remediation activities.

A row was added to the reduced input–output table to include data on direct sectoral wastewater discharges to wastewater tanks on Croatian territory without prior treatment (e.g., public sewers, watercourses, sea, soil, etc.) (see Figure A1). In order to achieve the highest possible reliability and credibility of the analysis, wastewater data from the same year to which the input–output table refers were considered. Quantities of direct wastewater discharges from industrial sectors were calculated based on data published in the CBS (Croatian Bureau of Statistics) statistical publication on water use and pollution control in industry in 2010 (see Table 1, Sectors 2–21) [73]. Wastewater quantities for all other sectors were estimated using data from the Croatian Environmental Pollution Register (ROO) [74] and the CBS statistical report Public sewage system, 2010 [75] (see Table A1).

In the assessment of direct wastewater discharges in the service sectors of the Croatian economy, only the hospitality industry was subdivided as a single sector, since it usually stands out as a significant wastewater discharger. Although a part of wastewater discharges

from households related to renting rooms and apartments can also be attributed to the hospitality sector, the quantities of these wastewater discharges are not considered in this study because there is no objective data base for their evaluation. Assuming that all service providers discharge their wastewater primarily into the public sewerage system, all other trade and service activities are grouped into a single *other services* sector.

Wastewater discharge data used in this study are expressed in cubic meters ($m^3$), while sectoral production values are expressed in Croatian kuna (HRK) based on monetary values in the official Croatian input–output table (according to the middle exchange rate of Croatian National Bank on 20 January 2022, 1 EUR = 7.52 HRK [76]). Therefore, the values of the calculated indicators of wastewater discharge intensity are expressed as a combination of these units of measurement.

It is important to emphasize that the wastewater discharge data used in this study correspond to what many other studies refer to as graywater discharges, conservatively assuming that water pollution in all production sectors has a dilution factor of 1 [77]. The graywater footprint is generally defined as the amount of freshwater required to assimilate pollutants to meet specified water quality standards [78]. Thus, assuming a dilution factor of 1 means that the volume of wastewater return flow discharged to the environment without prior treatment can be used as a surrogate measure of graywater. The main reason this assumption is often used in EEIO studies is that it is difficult to determine a dilution factor that represents sectors well at a higher level of aggregation due to their internal production diversity. In addition, the wastewater quality, i.e., the pollutant concentration in the wastewater discharges of the different sectors in the different countries, is not known in the literature. However, in the empirical part of this study, the original "untreated wastewater context" is used, so that the interpretation of the obtained results can be more understandable for a wider readership.

## 4. Empirical Results and Analysis

In accordance with the previously described methodology and the data sources used, in this section, we discuss an input–output analysis of wastewater discharge in the Croatian economy.

### 4.1. Intensity of Direct and Indirect Wastewater Flows in the Croatian Economy

Table 2 shows the sectoral structure of wastewater discharged directly into the environment of the Republic of Croatia, then the sectoral values of direct, indirect and cumulative wastewater discharge intensity and the values of indirect and cumulative multipliers of wastewater discharge in the Croatian economy.

According to the data in Table 2, the Croatian economy discharged a total of 96.47 million $m^3$ of wastewater into the environment, with the largest share of the discharged wastewater occurring in the industrial sectors (79.85%, sectors 2–21). The most significant water polluters in the Croatian economy are *the petroleum refining sector* and *the chemical sector*, which together accounted for 55.92% of the total direct wastewater discharge. *The hospitality sector* (9.36%) and *the food sector* (8.85%) also accounted for a significant share of direct wastewater discharges. Although *the agricultural* and *construction sectors* discharged relatively small amounts of wastewater compared to the aforementioned sectors, it is important to emphasize that they are one of the main sources of nonpoint source water pollution [79].

Comparing sectoral wastewater discharges per unit of output produced, it can be found that *the chemical* and *petroleum refining sectors* have significantly higher direct wastewater intensity compared to other sectors. The direct wastewater intensity of *the chemical sector* was 1802.13 $m^3$/million HRK, and in *the petroleum refining sector,* 1313.32 $m^3$/million HRK. *The water supply sector* recorded the third-highest direct wastewater intensity (763.05 $m^3$/million HRK), which indicates the ecological inefficiency of the water supply systems in the Republic of Croatia. In addition to the mentioned sectors, *the non-metallic sector* (458.36 $m^3$/million

HRK) and the *hospitality sector* (278.44 m$^3$/million HRK) also had relatively high direct wastewater intensity.

**Table 2.** Absolute and intensity indicators of wastewater discharge and wastewater multipliers (authors' calculation based on data shown in Figure A1).

| Sector Labels | Total Direct Wastewater Discharge (m$^3$) | % | Direct Wastewater Intensity (m$^3$/mil. HRK) | Indirect Wastewater Intensity (m$^3$/mil. HRK) | Cumulative Wastewater Intensity (m$^3$/mil. HRK) | Cumulative Wastewater Multiplier | Indirect Wastewater Multiplier |
|---|---|---|---|---|---|---|---|
| 1 | 2,596,000.00 | 2.69 | 94.11 | 265.92 | 360.04 | 3.83 | 2.83 |
| 2 | 2,041,000.00 | 2.12 | 101.11 | 36.05 | 137.17 | 1.36 | 0.36 |
| 3 | 8,540,000.00 | 8.85 | 205.76 | 172.7 | 378.47 | 1.84 | 0.84 |
| 4 | 1,300,000.00 | 1.35 | 95.3 | 171.97 | 267.27 | 2.8 | 1.8 |
| 5 | 436,000.00 | 0.45 | 115.54 | 137.48 | 253.03 | 2.19 | 1.19 |
| 6 | 1,065,000.00 | 1.1 | 171.43 | 102.64 | 274.08 | 1.6 | 0.6 |
| 7 | 93,000.00 | 0.1 | 27.76 | 265.38 | 293.15 | 10.56 | 9.56 |
| 8 | 25,411,000.00 | 26.34 | 1313.32 | 85.65 | 1398.96 | 1.07 | 0.07 |
| 9 | 28,536,000.00 | 29.58 | 1802.13 | 287.02 | 2089.16 | 1.16 | 0.16 |
| 10 | 57,000.00 | 0.06 | 7.14 | 88.19 | 95.33 | 13.35 | 12.35 |
| 11 | 95,000.00 | 0.1 | 13.97 | 127.72 | 141.68 | 10.14 | 9.14 |
| 12 | 3,332,000.00 | 3.45 | 458.36 | 153.41 | 611.77 | 1.33 | 0.33 |
| 13 | 981,000.00 | 1.02 | 54.37 | 50.57 | 104.93 | 1.93 | 0.93 |
| 14 | 222,000.00 | 0.23 | 10.99 | 48.96 | 59.95 | 5.46 | 4.46 |
| 15 | 139,000.00 | 0.14 | 12.25 | 25.52 | 37.77 | 3.08 | 2.08 |
| 16 | 744,000.00 | 0.77 | 46.41 | 43.38 | 89.79 | 1.93 | 0.93 |
| 17 | 293,000.00 | 0.3 | 41.71 | 87.07 | 128.78 | 3.09 | 2.09 |
| 18 | 66,000.00 | 0.07 | 7.9 | 167.06 | 174.96 | 22.14 | 21.14 |
| 19 | 1,167,000.00 | 1.21 | 85.59 | 317.05 | 402.64 | 4.7 | 3.7 |
| 20 | 2,361,000.00 | 2.45 | 763.05 | 119.32 | 882.36 | 1.16 | 0.16 |
| 21 | 148,000.00 | 0.15 | 25.42 | 268.44 | 293.86 | 11.56 | 10.56 |
| 22 | 286,000.00 | 0.3 | 5.9 | 140.49 | 146.39 | 24.8 | 23.8 |
| 23 | 9,034,000.00 | 9.36 | 278.44 | 124.32 | 402.76 | 1.45 | 0.45 |
| 24 | 7,525,000.00 | 7.8 | 23.24 | 79.4 | 102.64 | 4.42 | 3.42 |

The sectors with the highest direct wastewater intensity of production also generate the highest cumulative intensity of wastewater discharge per unit of production (see Figure 4).

From the quantities shown in Figure 4, it is clear that the Croatian economy pollutes the environment far less with wastewater generated from the production of inputs needed to produce a unit of output in the *chemical*, *petroleum refining*, *water supply*, *non-metallic* and *hospitality* sectors than with direct wastewater discharges in these sectors. This is best explained by the wastewater multipliers, whose values show the following (see Table 2):

- For every 1 m$^3$ of wastewater discharged in *the chemical sector*, the rest of the economy must discharge 0.16 m$^3$ of wastewater;
- For every 1 m$^3$ of wastewater discharged in *the petroleum refining sector*, the rest of the economy must discharge 0.07 m$^3$ of wastewater;
- For every 1 m$^3$ of wastewater discharged in *the water supply sector*, the rest of the economy must discharge 0.07 m$^3$ of wastewater;

- For every 1 m³ of wastewater discharged in *the non-metallic sector*, the rest of the economy must discharge 0.33 m³ of wastewater; and
- For every 1 m³ of wastewater discharged in *the hospitality sector*, the rest of the economy must discharge 0.45 m³ of wastewater.

The previously presented characteristics of the cumulative wastewater intensity of *the chemical* and *petroleum refining* sectors, as well as other sectors that dominate the structure of the total direct wastewater discharge in the Croatian economy, show that their discharges are largely determined by the sectors involved in the intermediate distribution of their production. In other words, the sectors that have the highest indirect wastewater intensity values have a significant impact on the wastewater load in the Croatian environment (see Figure 5).

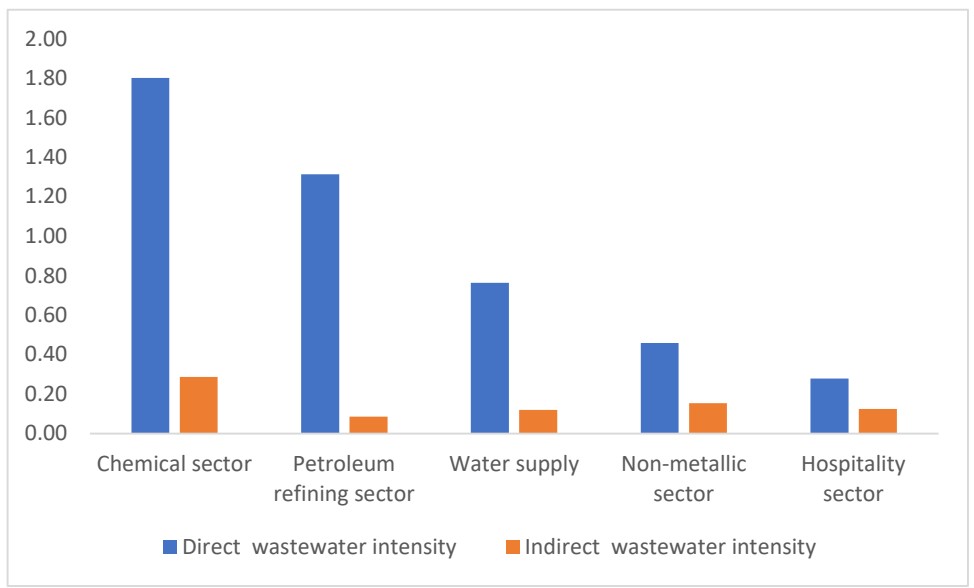

**Figure 4.** Structure of cumulative intensity of wastewater discharge in the main sectors (in 1000 m³/million HRK, made by authors based on data shown in Table 2).

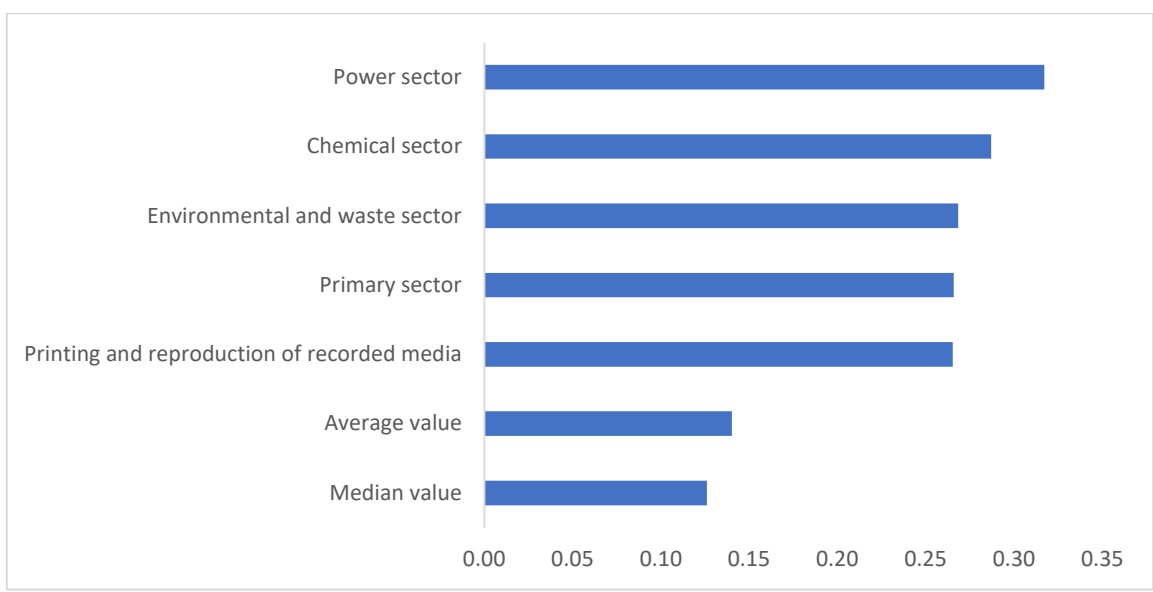

**Figure 5.** Sectors with the highest indirect wastewater discharge intensity (in 1000 m³/million HRK, made by authors based on data shown in Table 2).

According to the results of the model, *the power sector* has the highest indirect wastewater intensity (317.05 m$^3$/million HRK), and 83.86% of its indirect discharges are associated with the use of the output of the *petroleum refining sector* (see Figure A2). *The chemical sector* also has a high indirect wastewater intensity (287.02 m$^3$/million HRK). However, it is important to emphasize that most of the indirect discharges are due to the use of own products (88.56%), which de facto further increases the direct wastewater intensity of *the chemical sector*. The third largest indirect wastewater intensity in the Croatian economy was achieved by *the environmental and waste sector* (268.44 m$^3$/million HRK). This is because the products of the chemical industry are widely used in activities related to the cleaning and remediation of contaminated areas, the mitigation of environmental hazards and the reuse and disposal of waste. Therefore, 71.92% of the indirectly discharged wastewater generated per unit of production in *the environmental and waste sector* is due to the use of inputs from *the chemical sector* (see Figure A2). The results of the model show that the *primary sector* (265.92 m$^3$/million HRK) and *the printing and reproduction of recorded media sector* (265.38 m$^3$/million HRK) also have a relatively high indirect wastewater intensity.

Looking at the cross-sectoral flows of virtual wastewater (see Figure A2), it is clear that the differences between sectors in indirect wastewater intensity are primarily due to the different shares of *the chemical sector* and the *petroleum refining sector* in the structure of their intermediate inputs. On the other hand, although *the hospitality* and *food sectors* have a significant share in the total amount of wastewater discharged directly into the environment (see Table 2), most of their production is destined for final consumption, so these sectors are not so strongly represented in the indirect pollution of other production sectors in the Croatian economy. For example, more than half of the total output of *the chemical* and *petroleum refining sector* was spent on production in other sectors, while intermediate goods accounted for 13.31% of the total value of *food sector* output and 11.83% of the total value of *hospitality sector* output (see Figure A1).

In addition to the largest direct, indirect and cumulative sectoral wastewater intensities, it is necessary to determine which sectors have the greatest potential to generate wastewater discharge flows. The higher the share of indirect wastewater in the cumulative wastewater intensity of an individual sector, the greater its potential to trigger wastewater discharges in the Croatian economy, regardless of the current level of its direct and cumulative wastewater discharges. In other words, production sectors whose demand for production inputs is met through supply chains that are environmentally inefficient in terms of wastewater discharge have the greatest potential to increase it (see Figure 6).

The extent to which a rapid increase in production activities in the sectors shown in the previous graph can potentially influence the increase in total wastewater discharges in the Croatian economy can best be seen in the values of their wastewater multipliers (see Table 2):

- For every 1 m$^3$ of wastewater discharged in the *construction sector*, the rest of the economy must discharge 23.80 m$^3$ of wastewater;
- For every 1 m$^3$ of wastewater discharged in *the machinery and equipment repair and installation sector,* the rest of the economy must discharge 21.14 m$^3$ of wastewater;
- For every 1 m$^3$ of wastewater discharged in the *pharmaceutical sector*, the rest of the economy must discharge 12.35 m$^3$ of wastewater;
- For every 1 m$^3$ of wastewater discharged in *the environmental and waste sector*, the rest of the economy must discharge 10.56 m$^3$ of wastewater;
- For every 1 m$^3$ of wastewater discharged in the *printing and reproduction of recorded media sector*, the rest of the economy must discharge 9.56 m$^3$ of wastewater; and
- For every 1 m$^3$ of wastewater discharged in *the rubber and plastic products manufacturing sector*, the rest of the economy must discharge 9.14 m$^3$ of wastewater.

For example, if the multiplier effects of *the construction sector*, which has the highest value of the indirect wastewater multiplier, are decomposed, the *petroleum refining sector* must discharge 7.34 m$^3$ of wastewater for every 1 m$^3$ of wastewater directly discharged

in *the construction sector*, the *non-metallic sector* must discharge 6.38 m³ of wastewater, *the chemical sector* must discharge 4.67 m³ of wastewater, while all other sectors together must discharge 5.40 m³ of wastewater to meet the total direct and indirect input needs of *the construction sector*. On the other hand, using the matrix of technical coefficients of indirect wastewater discharge (Q), it is possible to break down the direct wastewater discharges according to the structure of the intermediate demand of each sector. For example, for every 1 m³ of wastewater discharged in *the machinery and equipment repair and installation*, *construction* and *power sectors*, *the petroleum refining sector* must discharge 13.04 m³, 7.34 m³ and 3.11 m³ of wastewater into the environment, respectively (see Figure A3).

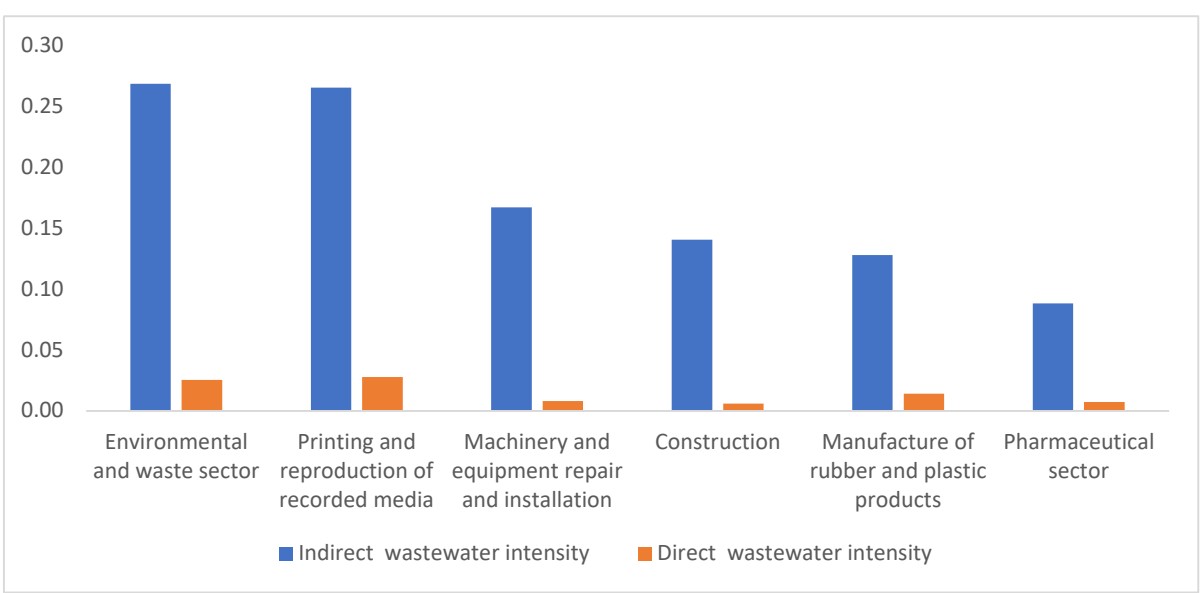

**Figure 6.** Sectors with the largest share of indirect intensity in the cumulative intensity of wastewater discharge (in 1000 m³/million HRK, made by authors based on data shown in Table 2).

*4.2. Sectoral Wastewater Footprints*

International trade is also an important factor in the production and allocation of national output, meaning that sectors indirectly contribute to the discharge of wastewater into the economies of other countries through the importation of goods and services, while their production for export contributes to the discharge of wastewater into the national economy. Therefore, to understand the extent to which wastewater pollutes the global environment to meet the demand for Croatian final goods and services, it is necessary to examine the flows of virtual wastewater in trade between Croatia and other countries. Using Equations (13)–(15) and the data in Figure A1, it is possible to calculate the wastewater footprint for all sectors of the Croatian production system (see Table 3).

The summarized sectoral results of the applied model show that the total wastewater footprint of Croatian producers is 85.2 million m³ of wastewater. To meet domestic demand, domestic producers discharged 67.9 million m³ of wastewater into the environment, while the net imported volume of virtual wastewater was 17.3 million m³. According to the calculated foreign trade balance of virtual wastewater, the wastewater footprint of export of the Croatian economy was 28.5 million m³ (i.e., the amount of indirect wastewater discharge of foreign producers into the Croatian environment), while the wastewater footprint of import was 45.8 million m³ (i.e., the amount of indirect wastewater discharge of domestic producers into the foreign environment).

**Table 3.** Sectoral wastewater footprints (in m³, authors' calculation based on data shown in Figure A1).

| Sector Labels | Domestic Wastewater Footprint | Net Imported Wastewater Footprint | Total Wastewater Footprint | Sector Labels | Domestic Wastewater Footprint | Net Imported Wastewater Footprint | Total Wastewater Footprint |
|---|---|---|---|---|---|---|---|
| 1 | 2,191,987.1 | 247,779.4 | 2,439,766.5 | 13 | 620,601.0 | 467,824.4 | 1,088,425.5 |
| 2 | 1,479,607.4 | 1,644,294.8 | 3,123,902.3 | 14 | 153,798.8 | 78,052.9 | 231,851.8 |
| 3 | 7,398,057.2 | 883,421.3 | 8,281,478.5 | 15 | 108,317.1 | 80,633.6 | 188,950.7 |
| 4 | 1,075,912.9 | 423,960.2 | 1,499,873.0 | 16 | 436,705.0 | 102,298.3 | 539,003.2 |
| 5 | 141,916.9 | −73,650.2 | 68,266.6 | 17 | 195,953.4 | 72,671.9 | 268,625.3 |
| 6 | 777,364.6 | 417,984.9 | 1,195,349.5 | 18 | 38,711.0 | −11,763.5 | 26,947.6 |
| 7 | 91,769.5 | 876.9 | 92,646.4 | 19 | 972,827.7 | 181,163.3 | 1,153,990.9 |
| 8 | 15,495,497.0 | −636,377.9 | 14,859,119.1 | 20 | 2,225,971.2 | −3138.7 | 2,222,832.5 |
| 9 | 16,528,705.7 | 12,714,475.5 | 29,243,181.2 | 21 | 93,455.6 | −36,820.0 | 56,635.6 |
| 10 | 39,692.3 | 18,182.4 | 57,874.7 | 22 | 277,632.6 | −5559.2 | 272,073.5 |
| 11 | 66,859.2 | 55,404.5 | 122,263.7 | 23 | 8,868,437.9 | 688,388.1 | 9,556,826.0 |
| 12 | 2,367,976.9 | 324,700.2 | 2,692,677.0 | 24 | 6,294,098.7 | −354,081.5 | 5,940,017.2 |

The following sectors have the largest share in the total domestic wastewater footprint of Croatia's gross production: *the chemical sector* (24.33%), *the petroleum refining sector* (22.81%), *the hospitality sector* (13.05%) and *the food sector* (10.89%). *The non-metallic* sector (3.49%), *the water supply sector* (3.28%) and *the primary sector* (3.23%) also have relatively large wastewater footprints. Except for *the primary sector*, most of the domestic wastewater footprint of all the above sectors relates to direct wastewater discharge, while about 74% of *the primary sector's* domestic wastewater footprint is indirect, i.e., through intermediate consumption (see Table 2, Figure A2).

Assuming that the intensity of wastewater discharge in domestic and foreign production is the same, Figure 7 shows the wastewater footprints of net imports and net exports of Croatian production sectors. Net importers have positive values and net exporters have negative values for the wastewater footprint of their international trade flows.

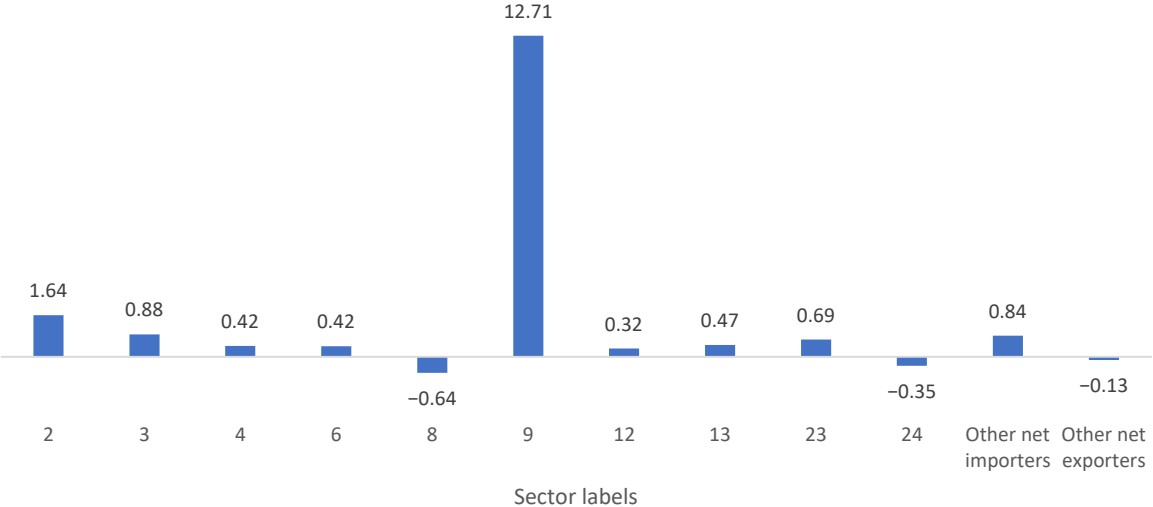

**Figure 7.** Net imported/exported wastewater footprints of Croatian production sectors (in 1,000,000 m³, made by authors based on data shown in Table 3).

By far the largest net importer of virtual wastewater is *the chemical sector* (12.7 million m³), while among the other sectors, only *mining and quarrying* has a wastewater footprint of net imports of more than 1 million m³ of wastewater (see Table 3). On the other hand, the largest net exporters of virtual wastewater in the Croatian economy are *the petroleum refining sector* and *the other services sector*, which together account for 88.3% of the total net exports of virtual wastewater (0.99 million m³, see Table 3). This indicates that the substitution of imports from *the chemical sector* with domestic production, while maintaining the same technological basis of production, can significantly increase wastewater discharge in the Croatian economy. In contrast, an increase in net exports from *the petroleum refining sector* and *other services sector* would not have such a large impact on the increase in wastewater discharges within the territorial boundaries of the Republic of Croatia.

By summing the sectoral domestic and net imported wastewater footprints, the structure of the total wastewater footprint of Croatia's gross production was determined, which is shown in Figure 8.

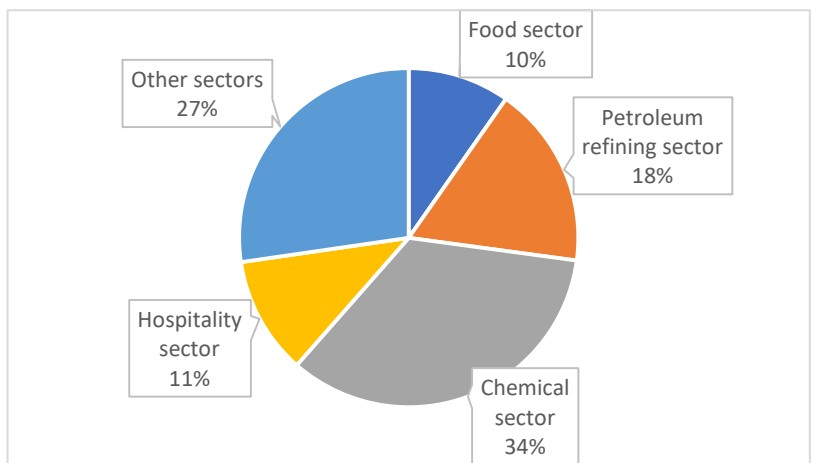

**Figure 8.** Structure of the total wastewater footprint of the Croatian production system by key sectors (made by authors based on data shown in Table 3).

In terms of wastewater pollution, the largest footprints were generated by *the chemical sector* (29.2 million m³), *the petroleum refining sector* (14.9 million m³), *the hospitality sector* (9.6 million m³) and *the food sector* (8.3 million m³). In relative terms, as much as 72.7% of the total wastewater footprint of national production is due to the use of the products and services of these sectors.

### 4.3. Sectoral Pull and Push Indices of Wastewater Discharge

To fully understand the impact of each sector on wastewater discharges in the Croatian economy, the relative strength of their forward and backward linkages in the process of gross production is analyzed below. Accordingly, the following table shows the values of the pull and push indices of wastewater discharges for all observed sectors.

According to the data presented in Table 4, the values of both indices for *the petroleum refining sector*, *the chemical sector*, *the non-metallic sector* and *the water supply sector* are greater than 1, which means that these sectors occupy a central position in the intermediate generation and discharge of wastewater in the Croatian economy. Therefore, a one-unit change in the demand for their final products and services leads to an above-average change in wastewater discharges in the Croatian economy. On the other hand, a one-unit change in the total demand for Croatian final products and services leads to an above-average change in wastewater discharges in the mentioned sectors. Among the other sectors, only *the power sector* and *the hospitality sector* have a pull index value of more than 1. This means that the process of procurement and consumption of intermediate goods and services in these sectors has an above-average impact on wastewater discharges in the Croatian economy. In

contrast, direct wastewater discharges in *the mining and quarrying* and *paper manufacturing sectors* are more responsive than average to a change in total final demand, which means that their wastewater-intensive production has a more pronounced forward strength.

**Table 4.** Sectoral pull and push indices of wastewater discharge (authors' calculation based on data shown in Figure A1).

| Sector Labels | Pull Indices | Push Indices | Sector Labels | Pull Indices | Push Indices |
|---|---|---|---|---|---|
| 1 | 0.95 | 0.50 | 13 | 0.28 | 0.29 |
| 2 | 0.36 | 2.96 | 14 | 0.16 | 0.11 |
| 3 | 1.00 | 0.47 | 15 | 0.10 | 0.09 |
| 4 | 0.70 | 0.17 | 16 | 0.24 | 0.08 |
| 5 | 0.67 | 0.23 | 17 | 0.34 | 0.10 |
| 6 | 0.72 | 0.80 | 18 | 0.46 | 0.43 |
| 7 | 0.77 | 0.04 | 19 | 1.06 | 6.15 |
| 8 | 3.68 | 5.02 | 20 | 2.32 | 0.42 |
| 9 | 5.49 | 3.85 | 21 | 0.77 | 0.27 |
| 10 | 0.25 | 0.16 | 22 | 0.38 | 0.04 |
| 11 | 0.37 | 0.37 | 23 | 1.06 | 0.29 |
| 12 | 1.61 | 0.91 | 24 | 0.27 | 0.22 |

## 5. Conclusions

In order to better understand the impact of intersectoral dependencies in the process of national production on the discharge of untreated wastewater of economic origin in the Republic of Croatia, in this study, we examined an extended input–output analysis of the Croatian economy. The analysis identified the characteristics of cumulative, direct and indirect wastewater intensity of the production sectors in the Croatian economy, as well as the multiplicative effects of their production activities on wastewater discharge. The analysis also included an assessment of the total, domestic and net imported wastewater footprint of Croatian production sectors, including the relative strength of their backward and forward linkages in terms of wastewater discharges.

The results of the analysis show that industrial activities are the main dischargers of wastewater in the Croatian economy, with the chemical and petroleum industries accounting for the largest share of total direct wastewater discharges. Since a significant part of the output of these industries is used by other sectors in the processes of intermediate consumption, the total volume of direct wastewater discharges from the chemical and petroleum refining sectors is largely determined by the share of other sectors in the structure of their intermediate supply and thus by changes in their final demand. The hospitality and food processing sectors are also important direct polluters of water resources, but these sectors are generally the last links in production value chains, so their wastewater discharges do not affect the intensity of indirect wastewater flows in the rest of the economy.

Because the results of this study are more positive rather than normative, they are not suitable for proposing precise policy measures to reduce wastewater discharges. However, the previously described cross-sectoral linkages in the Croatian economy show that it is not sufficient to use market-based instruments or regulations only for the sectors that account for the largest share of direct wastewater discharges. There is also a need to focus on sectors that have a large wastewater footprint due to their established supply chains. For example, encouraging investment in the construction of efficient wastewater treatment plants and improving management control and innovation capacity in industrial supply chains could significantly reduce the overall volume of untreated wastewater discharges in the Croatian

economy. This, of course, requires coordination between wastewater reduction and other public policies such as industrial, agricultural and trade policies.

The main limitation of this study concerns the degree of aggregation of the production sectors, which reduced the level of detail of the results obtained. This is mainly due to the lack of correspondence between the available monetary input–output data and the data on the discharge of untreated wastewater into the Croatian economy. Therefore, the production sectors in the model are determined by aggregating a number of related activities, which in reality do not have the same production technology and structure of production inputs. This is especially true for the other services sector. Moreover, due to data gaps for the analyzed year, the total direct discharges of untreated wastewater in the primary sector, construction, hospitality sector and other services were estimated using different sources, which also affected the quality of the obtained results.

When considering proposals for future research on direct and indirect economic impacts on water pollution in the Republic of Croatia, it is important to emphasize that industrial wastewater generally has a less favorable composition than wastewater from other sectors, as it contains higher concentrations of harmful and toxic substances such as organic and inorganic chemicals, chlorides, nitrogen and phosphorus compounds, heavy metals and oils. It is also important to consider that diffuse pollution, which is concentrated in water bodies through runoff from precipitation, also contributes significantly to water pollution. The problem with the flow of diffuse pollutants is that they are difficult to control in the environment. Aside from a small amount of stormwater captured by the sewer system, there are no effective techniques to capture and clean up diffuse pollutants. Therefore, future EEIO research on water pollution in the Croatian economy should focus more on material flows, i.e., flows of priority pollutants concentrated in the waters of the Republic of Croatia, while complying with EU water quality standards. Future research may also include complementing the EEIO with the LCA framework so that water pollution-based endpoint indicators can also be assessed in relation to human and ecosystem well-being in the Republic of Croatia.

**Author Contributions:** Conceptualization, S.Č.; methodology, S.Č.; formal analysis, S.Č.; investigation, S.Č., N.D.B. and A.J.; resources, S.Č., N.D.B. and A.J.; data curation, S.Č., N.D.B. and A.J.; writing—original draft preparation, S.Č., N.D.B. and A.J.; writing—review and editing, S.Č., N.D.B. and A.J.; visualization, S.Č.; supervision, S.Č. All authors have read and agreed to the published version of the manuscript.

**Funding:** This work was supported by the University of Rijeka (UNIRI) under the projects "Effect of exogenous changes on the impacts of nautical tourism" (no. ZIP UNIRI 116-2-21), "Smart cities in function of development of national economy" (no. uniri-drustv-18-255-1424) and "Impact of intangible capital in Croatian economy" (no. uniri-drustv-18-166).

**Institutional Review Board Statement:** Not applicable.

**Informed Consent Statement:** Not applicable.

**Data Availability Statement:** Links to the data sources considered in this study are provided in the Appendices A–D and References list.

**Conflicts of Interest:** The authors declare no conflict of interest.

## Appendix A

**Table A1.** Sources of collected data.

| Data | Source | Website Address and References |
|---|---|---|
| Discharge of untreated industrial wastewater in EU countries | Eurostat database | https://appsso.eurostat.ec.europa.eu/nui/show.do?dataset=env_ww_genv&lang=en [21] (accessed on 10 December 2021). |
| Gross value added in current prices of industry in EU economies | Eurostat database | http://appsso.eurostat.ec.europa.eu/nui/show.do?dataset=nama_10_a10 [22] (accessed on 10 December 2021). |
| Symmetric input–output table of the Croatian economy for 2010 | Croatian Bureau of Statistics | https://www.dzs.hr/Hrv_Eng/publication/2015/12-01-04_01_2015.htm [72] (accessed on 1 December 2021). |
| Untreated sectoral wastewater discharges in the Croatian economy | Utilization of waters and protection of waters from pollution in industry, 2010, Croatian Bureau of Statistics | https://www.dzs.hr/Hrv_Eng/publication/2011/06-01-01_01_2011.htm [73] (accessed on 1 December 2021). |
| | Public sewage system, 2010, Croatian Bureau of Statistics | https://www.dzs.hr/Hrv_Eng/publication/2011/06-01-06_01_2011.htm [75] (accessed on 1 December 2021). |
| | Croatian Environmental Pollution Register | http://roo.azo.hr/rpt.html [74] (accessed on 1 December 2021). |

## Appendix B

| Sector labels | 1 | 2 | 3 | 4 | 5 | 6 | 7 | 8 | 9 | 10 | 11 | 12 | 13 | 14 | 15 | 16 | 17 | 18 | 19 | 20 | 21 | 22 | 23 | 24 | Final demand (f) | Export (x) | Total output (X) |
|---|---|---|---|---|---|---|---|---|---|---|---|---|---|---|---|---|---|---|---|---|---|---|---|---|---|---|---|
| 1 | 4358.68 | 0.12 | 6562.56 | 164.27 | 325.87 | 148.68 | 46.55 | 0.03 | 161.08 | 123.24 | 122.40 | 5.78 | 2.15 | 0.31 | 1.23 | 28.65 | 141.23 | 2.18 | 14.29 | 0.91 | 24.03 | 43.37 | 697.28 | 2053.25 | 12,555.53 | 1.88 | 27,583.64 |
| 2 | 11.32 | 3105.10 | 1350.34 | 0.91 | 20.76 | 20.78 | 18.63 | 7406.10 | 97.85 | 23.07 | 24.89 | 242.09 | 58.24 | 6.81 | 14.34 | 3.72 | 25.16 | 540.37 | 1536.03 | 35.17 | 39.92 | 964.18 | 993.19 | 3265.17 | 380.84 | 0.37 | 20,184.97 |
| 3 | 888.36 | 0.06 | 972.56 | 153.86 | 15.41 | 16.96 | 3.80 | 0.07 | 129.37 | 34.87 | 18.82 | 2.79 | 0.60 | 0.30 | 0.20 | 0.34 | 35.76 | 0.37 | 1.92 | 5.78 | 36.43 | 22.14 | 2296.25 | 885.16 | 35,982.02 | 4.95 | 41,504.20 |
| 4 | 2.84 | 2.27 | 13.53 | 726.69 | 2.16 | 6.04 | 8.52 | 0.15 | 1.01 | 11.11 | 44.75 | 3.64 | 3.15 | 3.47 | 9.89 | 88.47 | 6.38 | 0.18 | 1.83 | 7.97 | 34.10 | 29.92 | 535.51 | 12,096.77 | 2.07 | 13,641.61 |
| 5 | 1.73 | 3.30 | 249.59 | 301.98 | 427.00 | 12.51 | 11.51 | 5.67 | 13.78 | 2.16 | 19.44 | 48.89 | 43.70 | 24.12 | 12.22 | 26.39 | 90.64 | 15.18 | 15.20 | 2.38 | 3.96 | 160.14 | 9.95 | 252.42 | 2019.65 | 3.71 | 3773.49 |
| 6 | 4.33 | 4.49 | 738.27 | 99.98 | 54.25 | 372.25 | 276.18 | 3.96 | 280.91 | 54.95 | 43.71 | 65.24 | 34.55 | 36.64 | 12.79 | 27.13 | 34.52 | 17.88 | 3.86 | 5.15 | 77.17 | 55.77 | 308.44 | 1780.57 | 1819.33 | 0.87 | 6212.34 |
| 7 | 0.00 | 0.16 | 19.23 | 3.48 | 0.77 | 33.93 | 60.45 | 0.24 | 3.02 | 1.62 | 3.52 | 1.63 | 1.53 | 1.38 | 0.71 | 3.97 | 1.33 | 1.61 | 1.68 | 0.40 | 0.71 | 0.87 | 1.71 | 104.00 | 3101.78 | 0.01 | 3349.73 |
| 8 | 468.99 | 104.14 | 355.86 | 24.46 | 9.35 | 8.16 | 9.50 | 239.83 | 33.67 | 3.90 | 5.88 | 65.48 | 47.47 | 23.00 | 7.19 | 8.71 | 8.92 | 519.10 | 2425.31 | 32.68 | 88.92 | 875.04 | 372.74 | 3946.97 | 9663.46 | 5.89 | 19,348.74 |
| 9 | 1987.85 | 20.19 | 969.40 | 764.66 | 72.21 | 139.85 | 263.44 | 42.57 | 1904.49 | 217.70 | 287.88 | 234.10 | 129.17 | 196.18 | 29.56 | 28.17 | 141.15 | 62.32 | 5.13 | 21.99 | 496.84 | 238.00 | 258.46 | 1208.83 | 6114.41 | 4.37 | 15,834.56 |
| 10 | 10.60 | 0.00 | 239.70 | 0.00 | 1.67 | 0.00 | 0.70 | 0.00 | 17.10 | 416.62 | 2.63 | 0.08 | 0.04 | 0.00 | 0.01 | 0.00 | 2.04 | 0.00 | 1.23 | 0.25 | 1.38 | 4.03 | 9.02 | 2972.15 | 4302.54 | 1.77 | 7981.78 |
| 11 | 35.50 | 34.69 | 998.01 | 169.22 | 45.00 | 39.72 | 358.49 | 9.88 | 120.01 | 104.18 | 455.81 | 604.66 | 107.92 | 217.44 | 88.82 | 279.45 | 96.97 | 131.66 | 6.66 | 35.97 | 47.18 | 529.85 | 147.30 | 1197.38 | 939.60 | 0.83 | 6801.41 |
| 12 | 30.99 | 15.06 | 248.32 | 34.49 | 21.58 | 0.43 | 2.33 | 5.84 | 22.75 | 23.91 | 6.63 | 214.40 | 58.26 | 91.79 | 20.31 | 53.71 | 17.59 | 47.93 | 10.79 | 82.28 | 16.03 | 3555.26 | 123.28 | 765.57 | 1799.93 | 1.61 | 7269.45 |
| 13 | 112.57 | 29.36 | 561.82 | 144.91 | 91.41 | 10.77 | 240.93 | 84.16 | 188.56 | 44.75 | 239.63 | 189.22 | 2132.86 | 822.76 | 399.44 | 922.39 | 387.08 | 405.27 | 21.51 | 67.40 | 460.56 | 2362.10 | 493.60 | 3429.70 | 4201.77 | 3.57 | 18,044.52 |
| 14 | 4.89 | 2.23 | 37.82 | 9.73 | 6.01 | 1.09 | 17.94 | 1.26 | 4.06 | 0.99 | 8.18 | 41.11 | 67.21 | 1620.98 | 97.38 | 390.40 | 65.75 | 480.67 | 51.01 | 25.73 | 9.45 | 517.03 | 28.56 | 1357.86 | 15,361.21 | 5.00 | 20,208.54 |
| 15 | 19.75 | 7.15 | 165.49 | 39.25 | 9.51 | 8.56 | 16.05 | 13.60 | 16.76 | 13.03 | 13.10 | 29.41 | 82.51 | 167.47 | 286.30 | 549.49 | 15.68 | 142.39 | 9.11 | 9.51 | 10.20 | 410.11 | 309.81 | 1502.58 | 7500.19 | 1.73 | 11,347.01 |
| 16 | 8.30 | 0.86 | 2.70 | 14.83 | 0.44 | 10.62 | 0.07 | 0.06 | 0.13 | 0.00 | 0.36 | 1.06 | 38.39 | 73.28 | 100.41 | 1342.62 | 2.54 | 175.24 | 0.35 | 5.19 | 122.27 | 132.44 | 4.51 | 1454.33 | 12,541.32 | 5.64 | 16,032.34 |
| 17 | 3.24 | 0.41 | 31.67 | 75.92 | 27.99 | 1.79 | 4.63 | 0.73 | 9.16 | 6.23 | 3.15 | 50.28 | 23.83 | 65.86 | 2.95 | 21.02 | 151.18 | 5.98 | 1.77 | 3.07 | 3.97 | 11.53 | 42.74 | 1636.55 | 4838.94 | 1.91 | 7024.58 |
| 18 | 689.46 | 14.57 | 285.93 | 33.01 | 17.35 | 33.94 | 26.95 | 36.29 | 80.57 | 23.60 | 15.61 | 45.01 | 143.99 | 172.74 | 75.38 | 125.66 | 18.39 | 325.37 | 234.80 | 111.83 | 59.27 | 214.55 | 312.31 | 2918.66 | 2337.40 | 2.33 | 8352.64 |
| 19 | 257.36 | 26.68 | 601.69 | 129.62 | 40.39 | 117.09 | 64.05 | 41.34 | 280.28 | 49.00 | 34.77 | 181.42 | 99.42 | 68.00 | 22.60 | 35.50 | 32.29 | 257.71 | 1050.17 | 91.42 | 133.49 | 511.72 | 1029.17 | 6880.36 | 1599.14 | 0.26 | 13,634.70 |
| 20 | 8.89 | 0.38 | 25.64 | 8.27 | 1.00 | 0.62 | 0.91 | 0.43 | 9.18 | 2.20 | 0.95 | 1.88 | 1.60 | 1.32 | 0.38 | 4.19 | 2.25 | 1.36 | 3.57 | 77.09 | 30.81 | 39.06 | 373.72 | 830.10 | 1668.40 | 0.00 | 3094.18 |
| 21 | 12.49 | 3.71 | 67.17 | 15.92 | 14.12 | 89.79 | 81.30 | 1.64 | 7.99 | 8.11 | 56.07 | 30.99 | 68.54 | 12.15 | 4.32 | 14.35 | 6.87 | 34.28 | 131.17 | 22.96 | 248.70 | 117.05 | 202.34 | 1343.22 | 3226.73 | 1.67 | 5822.00 |
| 22 | 18.19 | 5.46 | 102.34 | 20.76 | 8.94 | 4.94 | 6.76 | 7.24 | 18.17 | 6.43 | 5.48 | 28.99 | 18.69 | 13.75 | 6.00 | 8.20 | 6.51 | 20.80 | 25.76 | 73.99 | 53.04 | 2614.50 | 139.52 | 2223.61 | 43,011.89 | 0.87 | 48,449.96 |
| 23 | 47.88 | 0.38 | 27.18 | 3.90 | 1.05 | 2.29 | 0.66 | 0.54 | 1.18 | 1.64 | 0.94 | 2.37 | 3.37 | 2.53 | 1.67 | 5.14 | 1.79 | 3.13 | 4.03 | 3.82 | 13.84 | 50.53 | 231.71 | 3425.22 | 28,605.83 | 0.00 | 32,444.63 |
| 24 | 3116.68 | 756.87 | 6873.82 | 1458.46 | 442.23 | 626.09 | 678.39 | 1701.17 | 1180.86 | 600.55 | 514.65 | 1088.90 | 1617.41 | 1672.27 | 868.37 | 2060.36 | 668.26 | 1320.61 | 1682.33 | 572.57 | 905.38 | 15,492.03 | 4675.57 | 73,435.90 | 199,747.22 | 32.69 | 323,756.93 |
| Import (m) | 3.19 | 14.00 | 8.79 | 6.26 | 1.25 | 2.91 | 0.02 | 5.09 | 9.70 | 4.28 | 4.03 | 2.25 | 10.38 | 11.51 | 7.99 | 7.68 | 3.61 | 0.62 | 2.09 | 0.00 | 0.18 | 0.02 | 2.60 | 15.39 | | | |
| Total input (X) | 27,583.64 | 20,184.97 | 41,504.20 | 13,641.61 | 3773.49 | 6212.34 | 3349.72 | 19,348.73 | 15,834.56 | 7981.78 | 6801.41 | 7269.45 | 18,044.52 | 20,208.54 | 11,347.01 | 16,032.34 | 7024.58 | 8352.64 | 13,634.70 | 3094.18 | 5822.00 | 48,449.96 | 32,444.63 | 323,756.93 | | | |
| Direct wastewater discharges (w, in mil. m³) | 2.60 | 2.04 | 8.54 | 1.30 | 0.44 | 1.07 | 0.09 | 25.41 | 28.54 | 0.06 | 0.10 | 3.33 | 0.98 | 0.22 | 0.14 | 0.74 | 0.29 | 0.07 | 1.17 | 2.36 | 0.15 | 0.29 | 9.03 | 7.53 | | | |

**Figure A1.** Extended input–output table of the Republic of Croatia (24 × 24, in million HRK) with data on direct wastewater discharge (in million m$^3$, authors' calculation based on data described in Section 3.5).

## Appendix C

| Sector labels | 1 | 2 | 3 | 4 | 5 | 6 | 7 | 8 | 9 | 10 | 11 | 12 | 13 | 14 | 15 | 16 | 17 | 18 | 19 | 20 | 21 | 22 | 23 | 24 |
|---|---|---|---|---|---|---|---|---|---|---|---|---|---|---|---|---|---|---|---|---|---|---|---|---|
| 1 | 18.72 | 0.07 | 18.75 | 2.25 | 11.41 | 3.19 | 2.65 | 0.15 | 1.70 | 2.14 | 2.50 | 0.73 | 0.23 | 0.21 | 0.17 | 0.51 | 2.82 | 0.39 | 0.37 | 0.44 | 1.10 | 0.69 | 4.06 | 1.19 |
| 2 | 2.56 | 18.89 | 6.08 | 1.13 | 2.18 | 1.65 | 2.80 | 46.91 | 1.99 | 1.06 | 1.23 | 6.10 | 1.29 | 0.77 | 0.69 | 0.91 | 1.29 | 12.72 | 24.45 | 4.20 | 3.39 | 5.53 | 6.32 | 3.44 |
| 3 | 8.52 | 0.06 | 6.66 | 2.99 | 2.08 | 1.07 | 1.07 | 0.13 | 2.22 | 1.31 | 1.04 | 0.52 | 0.18 | 0.17 | 0.13 | 0.25 | 1.55 | 0.31 | 0.27 | 0.75 | 1.92 | 0.61 | 15.57 | 1.13 |
| 4 | 0.06 | 0.03 | 0.11 | 5.41 | 0.13 | 0.14 | 0.37 | 0.03 | 0.04 | 0.04 | 0.21 | 0.71 | 0.06 | 0.06 | 0.06 | 0.12 | 1.34 | 0.15 | 0.05 | 0.15 | 0.21 | 0.22 | 0.15 | 0.24 |
| 5 | 0.11 | 0.04 | 0.90 | 3.13 | 14.84 | 0.33 | 0.65 | 0.08 | 0.18 | 0.08 | 0.46 | 1.04 | 0.40 | 0.23 | 0.19 | 0.32 | 1.83 | 0.36 | 0.22 | 0.23 | 0.22 | 0.65 | 0.18 | 0.20 |
| 6 | 0.78 | 0.13 | 3.94 | 2.02 | 3.44 | 11.40 | 16.35 | 0.24 | 3.92 | 1.64 | 1.67 | 2.31 | 0.65 | 0.63 | 0.41 | 0.70 | 1.34 | 0.89 | 0.39 | 0.85 | 3.30 | 1.04 | 2.40 | 1.51 |
| 7 | 0.00 | 0.00 | 0.02 | 0.01 | 0.01 | 0.17 | 0.53 | 0.00 | 0.01 | 0.01 | 0.02 | 0.01 | 0.01 | 0.00 | 0.00 | 0.01 | 0.01 | 0.01 | 0.01 | 0.01 | 0.01 | 0.01 | 0.01 | 0.01 |
| 8 | 39.88 | 10.19 | 30.15 | 11.80 | 17.31 | 13.62 | 22.26 | 24.18 | 13.40 | 6.97 | 8.08 | 27.50 | 10.55 | 7.55 | 5.41 | 8.41 | 9.09 | 103.05 | 265.89 | 35.37 | 37.80 | 43.35 | 34.42 | 32.09 |
| 9 | 183.46 | 3.73 | 88.34 | 131.97 | 70.36 | 61.63 | 195.13 | 7.91 | 254.17 | 66.76 | 102.37 | 84.58 | 21.37 | 26.88 | 9.52 | 12.38 | 54.68 | 26.63 | 8.41 | 26.73 | 193.06 | 27.57 | 33.77 | 17.63 |
| 10 | 0.02 | 0.00 | 0.07 | 0.01 | 0.02 | 0.01 | 0.03 | 0.01 | 0.02 | 0.40 | 0.01 | 0.02 | 0.01 | 0.01 | 0.01 | 0.01 | 0.01 | 0.02 | 0.02 | 0.02 | 0.02 | 0.04 | 0.02 | 0.09 |
| 11 | 0.08 | 0.04 | 0.43 | 0.24 | 0.25 | 0.14 | 1.71 | 0.03 | 0.15 | 0.23 | 1.03 | 1.33 | 0.13 | 0.21 | 0.14 | 0.33 | 0.25 | 0.31 | 0.04 | 0.27 | 0.20 | 0.33 | 0.14 | 0.10 |
| 12 | 1.35 | 0.54 | 3.79 | 1.88 | 3.76 | 0.49 | 1.43 | 0.58 | 1.17 | 1.83 | 0.94 | 14.76 | 2.10 | 2.75 | 1.22 | 2.38 | 1.79 | 3.78 | 1.07 | 14.55 | 2.62 | 37.68 | 2.93 | 2.12 |
| 13 | 0.75 | 0.18 | 1.39 | 1.07 | 2.11 | 0.48 | 5.43 | 0.46 | 1.05 | 0.59 | 2.59 | 2.27 | 7.56 | 3.00 | 2.46 | 4.34 | 3.79 | 3.80 | 0.53 | 2.06 | 5.67 | 3.92 | 1.44 | 1.14 |
| 14 | 0.04 | 0.01 | 0.04 | 0.03 | 0.05 | 0.02 | 0.11 | 0.01 | 0.02 | 0.01 | 0.03 | 0.10 | 0.07 | 0.98 | 0.12 | 0.35 | 0.13 | 0.75 | 0.08 | 0.16 | 0.06 | 0.18 | 0.04 | 0.09 |
| 15 | 0.04 | 0.01 | 0.08 | 0.06 | 0.04 | 0.10 | 0.04 | 0.02 | 0.03 | 0.03 | 0.04 | 0.08 | 0.08 | 0.13 | 0.34 | 0.50 | 0.05 | 0.27 | 0.03 | 0.08 | 0.07 | 0.16 | 0.15 | 0.09 |
| 16 | 0.11 | 0.02 | 0.10 | 0.12 | 0.09 | 0.16 | 0.16 | 0.04 | 0.05 | 0.04 | 0.06 | 0.10 | 0.18 | 0.26 | 0.51 | 4.34 | 0.08 | 1.21 | 0.09 | 0.23 | 1.21 | 0.30 | 0.10 | 0.34 |
| 17 | 0.06 | 0.02 | 0.11 | 0.30 | 0.41 | 0.05 | 0.15 | 0.04 | 0.06 | 0.07 | 0.06 | 0.37 | 0.10 | 0.19 | 0.04 | 0.12 | 0.97 | 0.11 | 0.06 | 0.13 | 0.10 | 0.15 | 0.12 | 0.29 |
| 18 | 0.28 | 0.01 | 0.13 | 0.05 | 0.10 | 0.08 | 0.13 | 0.03 | 0.07 | 0.05 | 0.05 | 0.09 | 0.09 | 0.10 | 0.07 | 0.10 | 0.05 | 0.36 | 0.18 | 0.35 | 0.13 | 0.10 | 0.13 | 0.11 |
| 19 | 1.85 | 0.29 | 2.40 | 1.57 | 1.90 | 2.40 | 3.14 | 0.59 | 2.28 | 1.01 | 1.00 | 3.16 | 0.98 | 0.75 | 0.52 | 0.81 | 0.98 | 3.73 | 7.80 | 3.76 | 3.19 | 2.40 | 3.82 | 2.84 |
| 20 | 0.87 | 0.15 | 1.23 | 0.97 | 0.81 | 0.55 | 1.18 | 0.34 | 0.83 | 0.53 | 0.49 | 0.82 | 0.43 | 0.38 | 0.30 | 0.71 | 0.66 | 0.78 | 0.77 | 20.25 | 5.02 | 1.79 | 9.72 | 2.86 |
| 21 | 0.06 | 0.02 | 0.11 | 0.08 | 0.16 | 0.45 | 0.78 | 0.03 | 0.05 | 0.06 | 0.27 | 0.19 | 0.14 | 0.05 | 0.04 | 0.07 | 0.07 | 0.17 | 0.31 | 0.27 | 1.20 | 0.16 | 0.23 | 0.17 |
| 22 | 0.02 | 0.00 | 0.03 | 0.02 | 0.03 | 0.01 | 0.03 | 0.01 | 0.02 | 0.01 | 0.01 | 0.04 | 0.01 | 0.01 | 0.01 | 0.01 | 0.01 | 0.03 | 0.02 | 0.17 | 0.08 | 0.36 | 0.04 | 0.06 |
| 23 | 1.25 | 0.19 | 1.16 | 0.65 | 0.82 | 0.65 | 1.15 | 0.44 | 0.48 | 0.45 | 0.47 | 0.85 | 0.51 | 0.46 | 0.40 | 0.75 | 0.58 | 0.94 | 0.78 | 1.29 | 1.54 | 1.80 | 2.80 | 3.97 |
| 24 | 5.05 | 1.43 | 6.68 | 4.21 | 5.14 | 3.90 | 8.04 | 3.37 | 3.10 | 2.86 | 3.09 | 5.73 | 3.43 | 3.20 | 2.75 | 4.95 | 3.71 | 6.29 | 5.21 | 7.02 | 6.29 | 11.44 | 5.75 | 7.67 |

**Figure A2.** Matrix of intersectoral virtual wastewater flows (W, authors' calculation based on data shown in Figure A1).

## Appendix D

| Sector labels | 1 | 2 | 3 | 4 | 5 | 6 | 7 | 8 | 9 | 10 | 11 | 12 | 13 | 14 | 15 | 16 | 17 | 18 | 19 | 20 | 21 | 22 | 23 | 24 |
|---|---|---|---|---|---|---|---|---|---|---|---|---|---|---|---|---|---|---|---|---|---|---|---|---|
| 1 | 0.20 | 0.00 | 0.09 | 0.02 | 0.10 | 0.02 | 0.10 | 0.00 | 0.00 | 0.30 | 0.18 | 0.00 | 0.00 | 0.02 | 0.01 | 0.01 | 0.07 | 0.05 | 0.00 | 0.00 | 0.04 | 0.12 | 0.01 | 0.05 |
| 2 | 0.03 | 0.19 | 0.03 | 0.01 | 0.02 | 0.01 | 0.10 | 0.04 | 0.00 | 0.15 | 0.09 | 0.01 | 0.02 | 0.07 | 0.06 | 0.02 | 0.03 | 1.61 | 0.29 | 0.01 | 0.13 | 0.94 | 0.02 | 0.15 |
| 3 | 0.09 | 0.00 | 0.03 | 0.03 | 0.02 | 0.01 | 0.04 | 0.00 | 0.00 | 0.18 | 0.07 | 0.00 | 0.00 | 0.02 | 0.01 | 0.01 | 0.04 | 0.04 | 0.00 | 0.00 | 0.08 | 0.10 | 0.06 | 0.05 |
| 4 | 0.00 | 0.00 | 0.00 | 0.06 | 0.00 | 0.00 | 0.01 | 0.00 | 0.00 | 0.01 | 0.01 | 0.00 | 0.00 | 0.00 | 0.00 | 0.00 | 0.03 | 0.02 | 0.00 | 0.00 | 0.01 | 0.04 | 0.00 | 0.01 |
| 5 | 0.00 | 0.00 | 0.00 | 0.03 | 0.13 | 0.00 | 0.02 | 0.00 | 0.00 | 0.01 | 0.03 | 0.00 | 0.01 | 0.02 | 0.02 | 0.01 | 0.04 | 0.05 | 0.00 | 0.00 | 0.01 | 0.11 | 0.00 | 0.01 |
| 6 | 0.01 | 0.00 | 0.02 | 0.02 | 0.03 | 0.07 | 0.59 | 0.00 | 0.00 | 0.23 | 0.12 | 0.01 | 0.01 | 0.06 | 0.03 | 0.02 | 0.03 | 0.11 | 0.00 | 0.00 | 0.13 | 0.18 | 0.01 | 0.06 |
| 7 | 0.00 | 0.00 | 0.00 | 0.00 | 0.00 | 0.00 | 0.02 | 0.00 | 0.00 | 0.00 | 0.00 | 0.00 | 0.00 | 0.00 | 0.00 | 0.00 | 0.00 | 0.00 | 0.00 | 0.00 | 0.00 | 0.00 | 0.00 | 0.00 |
| 8 | 0.42 | 0.10 | 0.15 | 0.12 | 0.15 | 0.08 | 0.80 | 0.02 | 0.01 | 0.98 | 0.58 | 0.06 | 0.19 | 0.69 | 0.44 | 0.18 | 0.22 | 13.04 | 3.11 | 0.05 | 1.49 | 7.34 | 0.12 | 1.38 |
| 9 | 1.95 | 0.04 | 0.43 | 1.38 | 0.61 | 0.36 | 7.03 | 0.01 | 0.14 | 9.35 | 7.33 | 0.18 | 0.39 | 2.45 | 0.78 | 0.27 | 1.31 | 3.37 | 0.10 | 0.04 | 7.59 | 4.67 | 0.03 | 0.76 |
| 10 | 0.00 | 0.00 | 0.00 | 0.00 | 0.00 | 0.00 | 0.00 | 0.00 | 0.00 | 0.06 | 0.00 | 0.00 | 0.00 | 0.00 | 0.00 | 0.00 | 0.00 | 0.00 | 0.00 | 0.00 | 0.01 | 0.00 | 0.00 | 0.00 |
| 11 | 0.00 | 0.00 | 0.00 | 0.00 | 0.00 | 0.00 | 0.06 | 0.00 | 0.00 | 0.03 | 0.07 | 0.00 | 0.00 | 0.01 | 0.01 | 0.01 | 0.04 | 0.00 | 0.00 | 0.01 | 0.06 | 0.00 | 0.00 | 0.00 |
| 12 | 0.01 | 0.01 | 0.02 | 0.02 | 0.03 | 0.00 | 0.05 | 0.00 | 0.00 | 0.26 | 0.07 | 0.03 | 0.04 | 0.25 | 0.10 | 0.05 | 0.04 | 0.48 | 0.01 | 0.02 | 0.10 | 6.38 | 0.01 | 0.09 |
| 13 | 0.01 | 0.00 | 0.01 | 0.01 | 0.02 | 0.00 | 0.20 | 0.00 | 0.00 | 0.08 | 0.19 | 0.00 | 0.14 | 0.27 | 0.20 | 0.09 | 0.09 | 0.48 | 0.01 | 0.00 | 0.22 | 0.66 | 0.01 | 0.05 |
| 14 | 0.00 | 0.00 | 0.00 | 0.00 | 0.00 | 0.00 | 0.00 | 0.00 | 0.00 | 0.00 | 0.00 | 0.00 | 0.00 | 0.09 | 0.01 | 0.01 | 0.00 | 0.09 | 0.00 | 0.00 | 0.00 | 0.03 | 0.00 | 0.00 |
| 15 | 0.00 | 0.00 | 0.00 | 0.00 | 0.00 | 0.00 | 0.00 | 0.00 | 0.00 | 0.00 | 0.00 | 0.00 | 0.00 | 0.01 | 0.03 | 0.01 | 0.00 | 0.03 | 0.00 | 0.00 | 0.00 | 0.03 | 0.00 | 0.00 |
| 16 | 0.00 | 0.00 | 0.00 | 0.00 | 0.00 | 0.00 | 0.01 | 0.00 | 0.00 | 0.01 | 0.00 | 0.00 | 0.00 | 0.02 | 0.04 | 0.09 | 0.00 | 0.15 | 0.00 | 0.00 | 0.05 | 0.05 | 0.00 | 0.01 |
| 17 | 0.00 | 0.00 | 0.00 | 0.00 | 0.00 | 0.00 | 0.01 | 0.00 | 0.00 | 0.01 | 0.00 | 0.00 | 0.00 | 0.00 | 0.00 | 0.00 | 0.02 | 0.01 | 0.00 | 0.00 | 0.01 | 0.03 | 0.00 | 0.01 |
| 18 | 0.00 | 0.00 | 0.00 | 0.00 | 0.00 | 0.00 | 0.00 | 0.00 | 0.00 | 0.01 | 0.00 | 0.00 | 0.00 | 0.00 | 0.01 | 0.00 | 0.00 | 0.05 | 0.00 | 0.00 | 0.01 | 0.02 | 0.00 | 0.00 |
| 19 | 0.02 | 0.00 | 0.01 | 0.02 | 0.02 | 0.01 | 0.11 | 0.00 | 0.00 | 0.14 | 0.07 | 0.01 | 0.02 | 0.07 | 0.04 | 0.02 | 0.02 | 0.47 | 0.09 | 0.00 | 0.13 | 0.41 | 0.01 | 0.12 |
| 20 | 0.01 | 0.00 | 0.01 | 0.01 | 0.01 | 0.00 | 0.04 | 0.00 | 0.00 | 0.07 | 0.04 | 0.00 | 0.01 | 0.03 | 0.02 | 0.02 | 0.02 | 0.10 | 0.01 | 0.03 | 0.20 | 0.30 | 0.03 | 0.12 |
| 21 | 0.00 | 0.00 | 0.00 | 0.00 | 0.00 | 0.00 | 0.03 | 0.00 | 0.00 | 0.01 | 0.02 | 0.00 | 0.00 | 0.00 | 0.00 | 0.00 | 0.00 | 0.02 | 0.00 | 0.00 | 0.05 | 0.03 | 0.00 | 0.01 |
| 22 | 0.00 | 0.00 | 0.00 | 0.00 | 0.00 | 0.00 | 0.00 | 0.00 | 0.00 | 0.00 | 0.00 | 0.00 | 0.00 | 0.00 | 0.00 | 0.00 | 0.00 | 0.00 | 0.00 | 0.00 | 0.00 | 0.06 | 0.00 | 0.00 |
| 23 | 0.01 | 0.00 | 0.01 | 0.01 | 0.01 | 0.00 | 0.04 | 0.00 | 0.00 | 0.06 | 0.03 | 0.00 | 0.01 | 0.04 | 0.03 | 0.02 | 0.01 | 0.12 | 0.01 | 0.00 | 0.06 | 0.31 | 0.01 | 0.17 |
| 24 | 0.05 | 0.01 | 0.03 | 0.04 | 0.04 | 0.02 | 0.29 | 0.00 | 0.00 | 0.40 | 0.22 | 0.01 | 0.06 | 0.29 | 0.22 | 0.11 | 0.09 | 0.80 | 0.06 | 0.01 | 0.25 | 1.94 | 0.02 | 0.33 |

**Figure A3.** Matrix of technical coefficients of indirect wastewater discharge (Q, authors' calculation based on data shown in Figure A1).

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
