# Peer review of "Impact of Intersectoral Dependencies in National Production on Wastewater Discharges: An Extended Input–Output Study of the Croatian Economy"

_water, doi:10.3390/w14132122_

Round 1

Reviewer 1 Report

Thanks for dealing with my comments very carefully, I am this revision is good enough to publish.

Author Response

Dear Reviewer 1, 

Thank you for acknowledging our corrections and concluding that the manuscript is good enough for publication.

At the suggestion of another reviewer, we have added only two short sentences in the revised manuscript (lines 110-111 and 401-403). We hope that these changes are acceptable to you.

Best regards,
Authors

Reviewer 2 Report

Dear Authors,

You have an interesting article that should be published in Water after some clarifications that I have proposed in the comments section. I feel that you have applied well-aimed indicators to assess the production/wastewater relationship using the Croatian economy as a case study so I can suggest the following remarks to improve the readability of this manuscript:

Comment 1 - Introduction - I am missing an explanation why Croatia has so low level of treated industrial wastewater (26.9%?) what types of industrial activities stand for the untreated wastewater? Maybe you can give at least 3 biggest sources of untreated effluents.

Comment 2 - Figure 1. Can you explain why only central and eastern European countries were presented in Fig 1? 

Comment 3 - 3.3. Wastewater footprint - I see one issue with using the word "domestic" in some papers domestic means household/sanitary wastewater and in some "domestic" stands for national, so I would modify this wording to avoid misunderstanding. 

Comment 4 - Table 1 in Label 19 Power Sector do you also include heat production?

Comment 5 - General note: I would rather calculate the costs in HRK (Kunas) to EUR, while for an international audience HRK maybe not understandable. 

Author Response

This manuscript is a resubmission of an earlier submission. The following is a list of the peer review reports and author responses from that submission.

Round 1

Reviewer 1 Report

The paper lacks clarity, consistency and credibility. The authors should provide a definition of the term "virtual wastewater" and also clearly explain the methodology, i.e. how they followed the flows of outputs and inputs from/to different industrial sectors in order to quantify sectoral "virtual wastewater" and indirect wastewater intensity.

The introduction is too long. What is manufacturing wastewater (industrial wastewater)? What are energetic byproducts? What does productivity of wastewater mean? In introduction the authors should also state intended application of the results, be it for imposing charges or incentivizing.

Methodological framework is very detailed, but no clear relationship is shown later in the paper with its application to the data used and obtained.

Data that is more than 10 years old is used in EEIO model and that is inappropriate. The sources of data should be provided in descriptions of tables and figures.

What does the term "national environment" mean (chapter 4)? Why do figures 3 and 4 not consider the same sectors?

Reviewer 2 Report

Comments on the manuscript entitled "Impact of Intersectoral Dependencies in National Production on Wastewater Discharges: An Extended Input-Output Study of the Croatian Economy" submitted for possible publication in the Sustainability.

In general, I think that this manuscript does not provide a clear discussion about the main work as well as the contribution which should be distinguished from current studies.  

Comment 1: In the Abstract, readers expect a gap between research needs and literature, or the motivation would be hard to understand. Additionally, the key question should be declared in Abstract, Introduction, Discussion and Conclusion with different needs. In current version, authors did not present the gap, any new and attractive findings and suggestions.

Comment 2: In “Introduction” section, authors wasted too much effort on the less relevant background without well providing the arguments with a question-driven expression. Introduction section should provide the background knowledge of the research question, which should be carefully arranged with a full understanding on the importance of current issues and related studies. This manuscript, however, presented too much irrelevant or little relevant studies but less discussion on the key question. The discussion should be concise with more focus on the research gap to derive key question. I suggest that authors need to delete all of less relevant information and clearly discuss the gap between research needs and current studies with a well-arranged storyline, and rewrite the key question that can be explicitly distinguished from literature.

Comment 3: In the section “Overview of the literature”, authors did not arrange the discussion the with a clear logic, and thus it is difficult for readers to catch the key points from the review: what these the main gaps derived from previous studies, and how to full these gaps? Because there are too many studies applying extended input-output model to address the wastewater discharge, the explanation of motivation and key questions need to be given accurate understanding on the innovation points instead of listing less relevant information and big problems.

Comment 4: in the sections “Methodological framework”, several methods involved without clear explanation. Authors not only need to explain why several methods should be apply, but also need to provide the extension, modification and integration with a well-designed framework. I suggest that authors can provide a clear figure to show the framework integrated with all the methods applied in this study.

Comment 5: the conclusion section, the final suggestion given from this section is not very attractive, authors should reconsider the results with more concern on the policy needs and what other studies can learn from. Also policy implications need to derived from the discussion about the possibilities that how to understanding the findings, Please make a step forward to provide policy suggestion derived from the results and not just give some common-known lists.

Comment 6: the writing should be improved in whole structure because of too many redundant and excessive description. I suggest that authors can consider to rearrange the whole structure of the manuscript with a much clearer storyline to show out the key questions and their solutions based on more accurate literature analysis. The description and discussion about background information and basic knowledge should be more concise with more focus on the key questions.

Reviewer 3 Report

The study described the calculation of wastewater discharges of Croatia using input-output modeling. They show the main sectors generating waste water, inside the country as well as outside Croatia related to its imports. The paper is rather long, but pleasant to read though, describing in detail the methodology applied. I also appreciated the literature overview.

However, there are some serious issues at the basis of the assumptions. Although the paper aims to include all sectors of the economy, however, it only includes the point sources of waste water. Agriculture, a large polluter of diffuse waste water is left out.

But what I find most important is that if only the volumes of waste water are included, the quality of this waste water is not taken into account. For example, if a cubic meter of waste water from source A contains a concentration [x], and a cubic meter of waste water from source B 10 * [x], the waste water volumes are exactly the same. Moreover, if a pollutant, e.g. c, is far more dangerous than pollutant d, this is not accounted for. Using the methodology as applied in the paper therefore generates incompatible results. For example, figure 4 shows that the power sector generates the highest indirect waste water intensity, followed by the chemical sector. However, what do these numbers mean? The power sector probably generates more thermal pollution, the chemical sector more dangerous chemical pollution.

This difficulty has been included in the concept of the water footprint introducing the so termed grey water footprint. That indicator uses a dilution factor to account for the concentration and dangerous pollutant issues (see for example the water footprint manual from 2011). Not taking this perspective into consideration means a serious flaw of this analysis. Another flaw is the reference to Renault, 2003. That paper in report 12 in the UNESCO-IHE series reflects a discourse at an expert meeting introducing the water footprint concept. I could not find any reference to waste water discharge there. In the beginning of water footprint analyses only grey water footprints related to nitrogen were taken into account, something that is missing in this paper.

Given the issues mentioned above, I am rejecting this paper.

Some minor issues are:

Lines 51-52

Measures to reduce waste water discharge. I think it is about the quality of the discharge and not about absolute volumes.

Line 113. Here the public waste water system is introduced. The study needs to mention that this is the system boundary.

Figure 1. Here only eastern European countries and Turkey are shown. How about the western European countries?

Mention the source in the figure caption.

Methodological framework

This part is really long and based on existing literature. The authors could move this part to an appendix.

Line 350

Here equation 7 includes the waste water discharge (volume). See my comments above related to the value of this indicator.

3.3. Wastewater footprint

This part is the very weak point of the analysis. I suggest to include a grey water footprint rather than this one generation no information at all.

Table 1. Here the food sector is included, but the whole production chain, i.e. agriculture misses.

Conclusion

The conclusion section also includes a discussion that is not based on the results.

In a possible discussion section the weaknesses of this approach should be discussed. Like, for example, the use of the word footprint, that slightly relates to the footprint concept, but only assessed the volumes, rather than taking actual pollution into account.

Some references are incomplete, e.g. the use of et al.

Round 2

Reviewer 1 Report

The paper deals with the large amount of data, but the very research concept is insufficiently clear. Objectives and hypotheses should be clearly presented.

The introduction should be concise and precise, and the entire paper should be made more coherent, referring to and presenting only what is relevant.

The authors must substantiate 1. how and why they combine actual wastewater produced with the so-called virtual wastewater, 2. why they do not base their research on virtual water that is built in the products and services (since virtual wastewater can hardly be defined and tracked).

The labels of all the tables and figures must indicate the sources. It does not suffice to mention sources in the text only.

What is the source used for fig. 1 (exact link to 'productivity of industrial wastewater' data from reference 21 should be indicated).

What is the meaning of 'wastewater pollution in the Croatian economy', 'concentration of pollution' in the contexts used? 

The statement 'It is particularly important for the Republic of Croatia to understand the key drivers ..... ' - who should understand - stakeholders, the public, policymakers, polluters?

Reviewer 2 Report

Thanks authors for dealing with my comments carefully. The arguments are strong and acceptable, which made me reconsider my comments. I think my previous comments were not clear and maybe created some misunderstandings. I believe that authors have done a good work, but the writing could be better. I expected more discussion on scientific contribution for providing reference to other studies. The new comments can be found in the attachment.     

Author Response

Dear reviewer 2,

After your second review, it was clearer to us what you expected us to do to improve our manuscript. We have taken all your comments into account and corrected the text of the manuscript accordingly. Before reviewing the corrected version of the manuscript, please note the following:

1) Considering your suggestion that the manuscript should include a "well-designed framework" that clearly shows the relationship between the indicators used, we have decided to present it in the form of a figure at the end of the introduction, as we believe it fits best there.

2) Guided by your conclusion at the end of the last review - "But I also think it is OK that authors consider in a different way", we hope you will not hold it against us that we kept the original concept of literature review. Nonetheless, we have corrected some of the text in this chapter to improve the English language and to explain in more detail the literature gap that confirms the relevance of our study.

We thank you very much for your contribution and quality suggestions and hope that this latest version of the manuscript meets your criteria for publication in the journal.

Reviewer 3 Report

The authors made an effort to reply to all the comments on the first draft of the paper. However, there are only few changes compared to the original version. The replies on the specific comments are substantial though. I would suggest to include those comments in the paper itself putting everything in the proper perspective. 

Author Response

Dear reviewer 3,

In accordance with your recommendations, we have incorporated our main responses to your initial comments into the manuscript. All changes to the manuscript, including those requested by other reviewers and additional clarifications, are highlighted in red.

We thank you very much for your contribution and quality suggestions and hope that this latest version of the manuscript meets your criteria for publication in the journal.

Round 3

Reviewer 1 Report

 x